DOI: 10.1038/s41467-017-02176-x | OPEN

# Fluctuations in instantaneous frequency predict alpha amplitude during visual perception

Stephanie Nelli [1], Sirawaj Itthipuripat[1,2], Ramesh Srinivasan[3,4] & John T. Serences[1,5,6]

Rhythmic neural activity in the alpha band (8–13 Hz) is thought to have an important role in the selective processing of visual information. Typically, modulations in alpha amplitude and instantaneous frequency are thought to reflect independent mechanisms impacting dissociable aspects of visual information processing. However, in complex systems with interacting oscillators such as the brain, amplitude and frequency are mathematically dependent. Here, we record electroencephalography in human subjects and show that both alpha amplitude and instantaneous frequency predict behavioral performance in the same visual discrimination task. Consistent with a model of coupled oscillators, we show that fluctuations in instantaneous frequency predict alpha amplitude on a single trial basis, empirically demonstrating that these metrics are not independent. This interdependence suggests that changes in amplitude and instantaneous frequency reflect a common change in the excitatory and inhibitory neural activity that regulates alpha oscillations and visual information processing.

[1] Neurosciences Graduate Program, University of California, San Diego, CA, USA. [2] Learning Institute, King Mongkut's University of Technology Thonburi, 10140 Bangkok, Thailand. [3] Department of Cognitive Sciences, University of California, Irvine, CA, USA. [4] Department of Biomedical Engineering, University of California, Irvine, CA, USA. [5] Department of Psychology, University of California, San Diego, CA, USA. [6] Kavli Institute for Brain and Mind, University of California, San Diego, CA, USA. Correspondence and requests for materials should be addressed to S.N. (email: snelli@ucsd.edu) or to J.T.S. (email: jserences@ucsd.edu)

Encoding and transferring sensory information between neural ensembles relies on a balance of excitatory and inhibitory neural activity (E/I balance) that is reflected in ongoing oscillatory activity[1–16]. Many studies of information processing in visual cortex have focused on the role of oscillatory activity in the alpha band—a particularly prominent set of oscillations ranging from ~8–13 Hz. One theory, referred to here as the desynchronization account, holds that default alpha amplitude is relatively large in visual cortex, reflecting strong population-level synchronization and suppression of visual information processing. In contrast, when processing visual input, the E/I balance in relevant local circuits shifts, leading to a local desynchronization from the default rhythm and a subsequent reduction in alpha amplitude[13,17–23]. Consistent with this framework, high alpha amplitude is associated with reduced perceptual sensitivity, presumably owing to a failure of relevant local circuits to desynchronize from the default rhythm[24–26]. Furthermore, alpha amplitude modulations track the relevance of stimuli in a topographically selective manner: spatial attention decreases amplitude in areas of visual cortex encoding attended regions of the visual field and increases amplitude in areas encoding task-irrelevant regions[21,27–34]. Finally, the relatively slow time-scale of these amplitude modulations ( > 100 ms) suggests correspondingly slow alterations between periods of efficient and inefficient visual information processing (for review see ref. [17]).

Although the desynchronization hypothesis focuses on relatively slow changes in alpha amplitude, rapid, cycle-by-cycle fluctuations in alpha oscillations are also thought to reflect alterations in the E/I balance and hence the efficacy of visual information processing[14,24–26,35–39]. This account, referred to here as the instantaneous frequency account, posits that epochs of neural excitability and efficient visual information processing are associated with a particular phase of ongoing alpha oscillations. These shorter and more rapidly occurring alternations in the E/I balance are thought to enhance perception both by sharpening feature tuning to stimuli and by temporally concentrating neural activity, thereby increasing the probability that activity is propogated to downstream areas[3,8,21,40–43]. Consistent with this account, a recent report suggests that instantaneous alpha frequency reflects the temporal density of periods of maximal perceptual sensitivity and the rate at which visual information is sampled and processed[35]. Thus, similar to the desynchronization account, the instantaneous frequency account also holds that alpha oscillations index changes in the E/I balance and the efficiency of information processing. However, the transitions between information processing states indexed by instantaneous frequency are theoretically linked to changes in the sampling rate of the visual system, and occur on a finer temporal scale than the more sustained transitions associated with alpha amplitude modulations.

As outlined above, alpha amplitude ($A$) and instantaneous frequency ($\omega$) are typically assumed to reflect independent processes, meaning that a sinusoidal voltage measurement ($V$) from a neural region at time $t$ can be described simply with $V(t) = A \sin(\omega t)$. However, work in mathematics and dynamical theory suggests that these assumptions may be an over simplification, especially in complex systems like the brain (for review see ref. [44]). Instead, interactions between the oscillations in driving and target neural regions should give rise to interdependencies between amplitude and frequency. As a simple analogy, imagine jumping on a trampoline with a partner jumping at very dissimilar rate, or frequency. In this case, the height, or amplitude, of your jumps will be relatively low. As your partner changes the frequency (and phase) of their jumps to match yours, the amplitude of your jumps will increase (a situation referred to as

resonance). However, even with maximal resonance you cannot jump infinitely high because of other factors such as air resistance and the finite stretchiness of the trampoline, forces that act as damping mechanisms. Although not a perfect analogy, this conceptual framework can serve as a starting point to understand interactions between the amplitude and instantaneous frequency of cortical responses in the alpha band. Here, we sought to first articulate the formal relationship between frequency and amplitude, and then to empirically test the proposed relationship using EEG. Our results suggest that amplitude and frequency are linked, and thus both metrics likely reflect the operation of a common dynamical system involved in determining the efficiency of visual information processing.

## Results

**Linking amplitude and frequency**. Amplitude and frequency are often discussed as independent metrics, although in complex systems they can be tightly coupled. Consider two interacting neural ensembles that naturally oscillate at different characteristic frequencies, such as might be observed in the thalamo-cortical or cortico-cortical circuits that give rise to alpha oscillations[45,46]. Here, we discuss coupled harmonic oscillators for simplicity, although models involving detailed biophysics exist[44,47,48]. First, let the uncoupled driving and target regions oscillate at characteristic frequencies $\omega_D$ and $\omega_T$, which themselves depend on connectivity and local E/I activity[49,50]. When considered as a coupled system, alpha amplitude in the target region ($A_T$) will be a function ($f$) of both the amplitude of the oscillatory drive ($A_D$) and the difference between the frequency of the driving and target oscillator, or $A_T = A_D * f(\omega_T - \omega_D)$ (See Supplemental Methods for model and derivation). In addition, the neural oscillations evoked by stimuli are transient (i.e., damped), making neurons sensitive to fine temporal structure in sensory inputs or inputs from other neuronal populations[48]. Interestingly, the damping mechanisms regulating the oscillatory response to these inputs (for example: leak conductance, capacitance, and voltage-gated currents[8]) will also modulate the effective characteristic frequency in the target region ($\omega_{eT}$)[8,50]. This means that the effective characteristic frequency in the target region is bounded by the theoretical characteristic frequency ($\omega_T \geq \omega_{eT}$). Substituting into the above statement, we now have $A_T = A_D * f(\omega_{eT} - \omega_D)$ (Supplemental Methods). This potential dependence complicates the traditional interpretation of alpha amplitude and instead suggests that shifts in amplitude reflect changes in the instantaneous frequency of the underlying dynamical system, which could arise given changes in oscillatory drive ($\omega_D$), local dampening ($\omega_{eT}$), or local characteristic frequency ($\omega_T$; Supplemental Methods).

The amplitude spectrum of typical EEG signals recorded over visual cortex shows a pronounced and focal bump centered on the dominant alpha frequency (Fig. 1b). This focal alpha bump is thought to be the result of resonant responses between interacting neural oscillators[8,40,50] (note the similarity to Supplementary Figure 1a). Thus, we hypothesized that the frequency-amplitude relationship outlined above is reflected in each subject's alpha bump. We expect that changes in the instantaneous frequency will lead to changes in amplitude, and that the precise nature of these changes will be captured by the shape of each subject's spectrum (Fig. 1a–c, Supplementary Figure 2b).

**Visual discrimination**. To test the potential interdependence between instantaneous frequency and amplitude, we designed a task in which subjects reported whether a low-contrast Gabor presented for ~8.3 ms was horizontal or vertical (two alternative-forced-choice orientation discrimination task, Fig. 1d, referred to

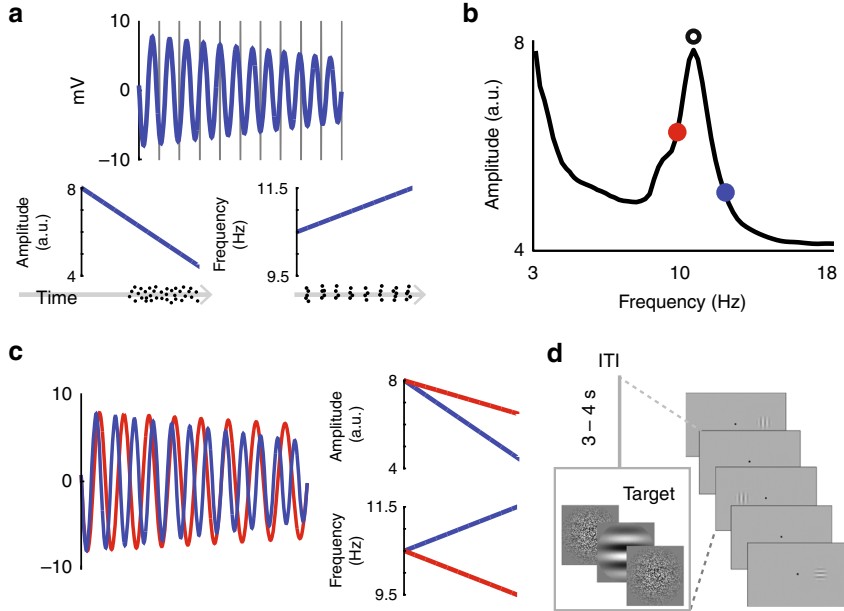

**Fig. 1** Hypothesis and task design. **a** A simulated example of an alpha oscillation that is both increasing in frequency and decreasing in amplitude over time, as exemplified in the left and right plots underneath, respectively. Verticle lines indicate evenly spaced time bins matching one cycle of the initial oscillatory frequency. Plotted below the amplitude and frequency traces are hypothetical raster plots corresponding with periods of efficient visual information processing according to the desychonization and instantaneous frequency hypotheses, respectively. **b** Amplitude spectrum from a representative subject. Note the general 1/f distribution of amplitude over frequency, and the pronounced bump in the alpha range. The circular outline indicates peak alpha frequency, whereas gray dots indicate hypothetical shifts away from the peak alpha frequency over the course of the trials outlined in **c**. **c** Along with the same example trial in **b**, now termed a correct trial, we have plotted a hypothetical incorrect trial that decreases in frequency and amplitude with magnitudes corresponding with the spectrum in **c**. Note that on the left side of the panel, the two traces are in phase, but become out of phase over the course of the trial, meaning frequency shifts could lead to offsets in phase through a relative speeding or slowing of the underlying signals. In addition to phase offsets, shifts in frequency away from peak alpha could also impact alpha amplitude as shown in the bottom right panel. **d** Task Design. The target was a Gaussian—windowed Gabor (mean contrast = 5%) presented for 8.3 ms. The target was immediately preceded and followed by one frame (~8.3 ms each) of gaussian—windowed white noise. Between target presentations, subjects passively fixated at the center of a gray screen for 3000–4000 ms (uniform distribution of ITIs). Target location (centered 8.5° left or right from fixation) was randomly selected with the only constraint that an equal number of trials were presented on both sides of fixation

as Experiment 1). A target Gabor could be presented on either the left or right side of the screen with a variable interval of 3000–4000 ms separating presentations. Performance during EEG recording was carefully titrated to 65% ( ± 2.8% SD) to obtain enough incorrect trials. Mean reaction time was 1106 ms, with faster RTs for correct (1073 ms) as compared with incorrect (1170 ms) trials (paired $t$-test $t(15) = −6.33$, $p < 0.0001$). Finally, subjects performed equally well on trials with vertical and horizontal targets and displayed no bias toward targets presented on one side of the screen (paired $t$-test, both $t(15)$'s < 0.87, $p$'s > 0.4).

**Characterizing ERPs alpha amplitude and alpha frequency**. Before directly assessing the potential link between alpha amplitude and instantaneous frequency, we first make contact with similar experimental paradigms by replicating event-related potential (ERP), alpha amplitude, and instantaneous frequency results from electrode groups contralateral and ipsilateral to the target location (Fig. 2a, Methods). We characterized task-related modulations of two ERP components evident in the grand average waveforms: an early negative deflection thought to index sensory processing and attentional selection[51–57], and a central-parietal late positive deflection thought to index post-sensory decision-related processing (e.g., decision difficulty, speed, and confidence)[54,58–61]. The early negative deflection (210–260 ms post-stimulus, see Methods) was significantly larger in electrodes contralateral compared to ipsilateral to the target ($t(15) = −3.4$, $p = 0.0001$), and showed a significant interaction between electrode

location and behavioral performance ($t(15) = −3.1$, $p = 0.01$, Fig. 2b, c; Table 1). The late positive deflection (460–510 ms post stimulus, see Methods) was larger on correct compared with incorrect trials ($t(15) = 6.5$, $p = 0.0$), and higher amplitude in contralateral compared to ipsilateral electrodes ($t(15) = 2.1$, $p = 0.0499$, Fig. 2b, c; Table 1). Note that the slightly delayed peaks of our ERP components are consistent with the low-contrast of our stimulus and difficulty of our task[25,54,62]. Together, these results suggest that sensory representations were topographically selective and that decision processes were impaired on incorrect trials.

Next, we examined whether modulations in alpha amplitude predicted behavioral performance. Many previous studies used attentional cues and analyzed anticipatory, pre-stimulus decreases in alpha amplitude[24,31,63]. However, as there was no advanced information concerning target location or timing in our paradigm, we expected to find amplitude decreases only after the stimulus (for review see ref. [64]). Consistent with previous reports, the average magnitude of post-stimulus alpha amplitude decreases depended on both behavioral accuracy and electrode location such that there were larger decreases on correct trials and in contralateral electrodes compared with ipsilateral electrodes (leading to an interaction between behavioral performance and electrode location; Fig. 3a; [17,26,33,64,65]). These amplitude decreases are consistent with the desynchronization account that decreases in alpha amplitude reflect a desynchronization of local alpha rhythms from a state that impairs visual information processing.

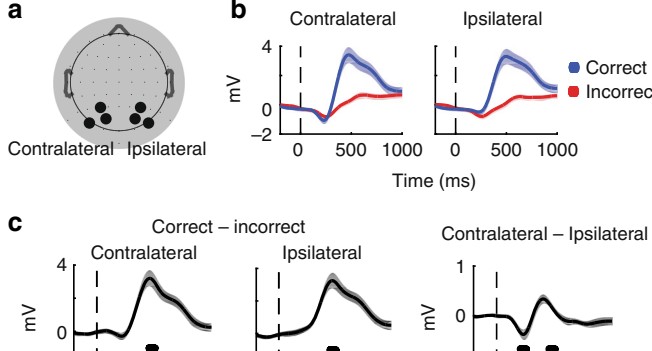

**Fig. 2** Event-related potentials confirm involvement of perceptual processes. **a** ERPs on correct (blue) and incorrect (red) trials in the contralateral and ipsilateral electrodes indicated in the topography plot to the right in panel **b** (note that electrode labels are flipped accordingly so that, by convention, electrodes contralateral to the target are shown on the left, see Methods). Dashed vertical line indicates target onset. **b** Topography for contralateral and ipsilateral electrodes used for all analyses are outlined on the 64 electrode Biosemi electrode scheme used in these experiments. **c** Difference waves between correct and incorrect trials. The END is more negative in contralateral than ipsilateral electrodes, resulting in a significant interaction between behavioral performance (correct vs. incorrect) and electrode location (contralateral vs. ipsilateral). Most significantly, there is a sustained increase in LPD amplitude on correct as compared with incorrect trials in both contralateral and ipsilateral electrodes. Dots below waveforms indicate a significance difference from zero as obtained from resampled t-tests performed on average amplitudes within the 50 ms time windows indicated by the dot width (210–260 and 460–510 ms post stimulus for the END and LPD, respectively). Significant main effects are indicated in black while purple indicates a significant interaction, all at $P = 0.05$

We then assessed whether higher instantaneous alpha frequency results in enhanced sensitivity to incoming visual information, as recently reported by Samaha & Postle 2015[35] (for instantaneous frequency derivation see Methods and ref. [66]). On average, we found significantly faster pre-stimulus instantaneous alpha frequency on correct trials, but only in contralateral electrodes ($t(15) = 3.4$, $p = 0.0008$, Fig. 4a, b; Table 2). These pre-stimulus shifts in instantaneous frequency may reflect a voluntary process of preparing for target processing. However, we cannot rule out the possibility that these shifts reflect spontaneous fluctuations because we did not use a pre-cue and we *post hoc* sorted the trials based on behavioral performance. In either case, this pattern of results is consistent with the hypothesis that increases in instantaneous alpha frequency in regions processing relevant information correspond to more efficient sampling and processing of visual information. Indeed, instantaneous frequency shifts of similar magnitudes have been reported to impact the effective resolution of visual perception[35] and spike timing in biophysical models[66].

**Predicting alpha amplitude**. As shown in Fig. 4c, instantaneous alpha frequency is highly dynamic and fluctuates by $5.93 \pm 0.64$ Hz over the course of single trials (mean ± SD, Fig. 4c). To test the hypothesis that task-related instantaneous frequency shifts result in concurrent modulations in alpha amplitude, we used instantaneous frequency to index into amplitude spectra for

each subject and electrode. This analysis effectively treats the spectra as look-up-tables to generate predicted alpha amplitudes (PaA) for each timepoint and trial (Fig. 1b, Supplementary Figure 2, see Methods).

If the amplitude spectrum is a valid transformation between instantaneous frequency and amplitude, PaA modulations should track measured amplitude modulations. Indeed, average PaA on correct and incorrect trials resembled measured modulations in alpha amplitude (Fig. 5a, Fig. 3a). Post-stimulus decreases in PaA depended on accuracy and electrode location in a manner similar, although not identical, to alpha amplitude (Fig. 5b, Table 2). As we were interested with the relationship between instantaneous frequency and amplitude within single trials, we computed timepoint-by-timepoint correlations between PaA and amplitude across all trials and found a significant relationship (mean correlation = $0.4773 \pm 0.0732$ SD, $p = 0$, Table 3, Fig. 6a; left panels, see Methods). These correlations were stable over time, and did not depend on behavioral performance or the position of the electrode with respect to the target (p-values do not survive FDR correction, Fig. 5a; right panels, Supplementary Table 1). Finally, modulations in PaA on single trials closely tracked those observed in amplitude (Fig. 5b).

To further investigate how frequency, amplitude, and behavior are related via the non-monotonic shape of the amplitude alpha spectra, we next sorted trials into four bins based on average instantaneous frequency in pre-stimulus and post-stimulus epochs that were significant in the analyses presented in Figs. 3b and 4b (see Methods). We binned trials based on whether average frequency was much lower than peak alpha, lower than peak alpha, greater than peak alpha, or much greater than peak alpha (quartile split). We then averaged pre or post-stimulus alpha amplitude and PaA over the trials in each of these bins. We observed a clear inverted-U relationship between amplitude and instantaneous frequency in both pre-stimulus and post-stimulus epochs (Supplementary Table 2, Fig. 6c). This is consistent with the main analysis showing that each subject's alpha spectrum maps changes in instantaneous frequency onto changes in alpha amplitude (see Fig. 1). Furthermore, only $25.1 \pm 5\%$ SD of trials in each pre-stimulus bin were still in that bin in the post-stimulus epoch, again emphasizing the dynamic changes in frequency that occur across single trials (25% is expected purely by chance).

Together, these analyses show that amplitude modulations were accurately predicted by passing instantaneous frequency through amplitude look-up-tables, evidenced by similar average PaA and amplitude waveforms, significant timepoint-by-timepoint PaA—amplitude correlations, and similar modulations of PaA and amplitude over single trials.

As both instantaneous frequency and amplitude are computed from bandpass filtered EEG data, we next addressed the concern that PaA-amplitude correlations were an artifact of filtering by passing instantaneous frequency through 5000 randomly generated white noise look-up-tables to generate $PaA_{Noise}$ (see Methods). This analysis yielded correlations between $PaA_{Noise}$ and actual alpha amplitude that were close to 0 (mean correlation = $3.4 \times 10^{-21} \pm 8.1 \times 10^{-16}$ SD, Supplementary Figure 3, Table 3). We next evaluated the empirical probability of observing the PaA—amplitude correlations reported in Fig. 6 under the null hypothesis of no relationship between these factors. To do this, we shuffled the frequency axis of each look-up-table, and passed instantaneous frequency through these shuffled look-up-tables to generate $PaA_{Shuff}$ a process we repeated 5000 times for each subject and electrode (see Methods). Average correlations between $PaA_{Shuff}$ and amplitude were ~24× smaller than those empirically observed, and P-values computed by comparing observed correlations to the $PaA_{Shuff}$ correlations were all equal to zero (mean correlation = $-0.0178 \pm 0.1265$ SD, Table 3,

**Table 1 Analysis of the early negative and the late positive event-related potentials**

|  | Early negative deflection | Late positive deflection |
|---|---|---|
| Contralateral vs ipsilateral | *$t(15) = -3.435$, $p = 0.0001$ | *$t(15) = 2.129$, $p = 0.0499$ |
| Accuracy, contralateral electrodes | $t = -1.745$, $p = 0.0984$ | *$i(15) = 6.548$, $p = 0.0$ |
| Accuracy, ipsilateral electrodes | $t = 1.258$, $p = 0.2281$ | *$t(15) = 7.028$, $p = 0.0$ |
| Location×accuracy interaction | *$t(15) = -3.053$, $p = 0.01$ | $t(15) = -1.244$, $p = 0.2301$ |

First, data were analyzed as a function of the location of electrodes with respect to the target (i.e., the amplitude of ERP responses in electrodes that were contralateral or ipsilateral to the target). Next, comparisons were made between correct and incorrect trials, separately for contralateral and ipsilateral electrodes. Finally, the interaction between electrode position (contralateral/ipsilateral) and behavioral accuracy was assessed. Note that all statistical tests are reported as t-tests on difference scores instead of F-values that would be obtained in an analysis of variance (ANOVA). This was done to maintain consistency across comparisons, and produces identical outcomes ($t$ is the square root of $F$ in this situation). All tests report t-tests on the average amplitude values within pre-defined 50 ms windows from 210–260 and 460–510 ms post stimulus for the END and LPD, respectively. t-values were compared against distributions obtained empirically by randomizing condition labels 10,000 times and then repeating the same statistical test (see Methods). * indicates a significant effect at $p = 0.05$

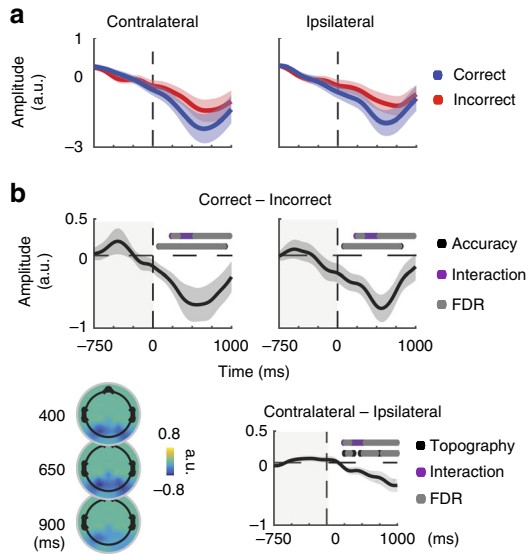

**Fig. 3** Topographically selective increases in post-stimulus amplitude predict accuracy. **a** The timecourse of alpha amplitude on correct (blue) and incorrect (red) trials in the contralateral and ipsilateral electrodes indicated in Fig. 1. Amplitude timecourses are baselined to −1000 to −750 ms pre-stimulus, and shaded areas indicate ±1 SEM within subject. **b** Alpha amplitude decreases more on correct compared to incorrect trials in both contralateral and ipsilateral electrodes. Furthermore, the decrease in alpha amplitude is topographically selective, displaying larger decreases contralateral to the target. Topographic plots indicate the difference between correct and incorrect trials averaged over 100 ms bins centered on 0.4, 0.65, and 0.9 s after stimulus onset. All dots indicate significance from zero, evaluated by comparing the obtained t-value with a null distribution of t-values computed by shuffling the condition labels 10,000 times. This analysis was done on a timepoint-by-timepoint basis from stimulus onset to +1 s, as indicated by the non-shaded areas (see Methods). Main effects with $P < 0.05$ are indicated in black, and gray dots indicate significance after FDR correction at $P = 0.05$

Supplementary Figure 3). Finally, to evaluate whether our results are specific to the unique shape of the resonant alpha bump, we fit a two-term exponential model to each spectrum, generating a new set of look-up-tables that captured only the 1/f falloff and not the alpha bump (see Methods). We then passed instantaneous frequency through these new look-up-tables to generate PaA$_{1/f}$. Again, PaA$_{1/f}$ was weakly correlated with amplitude (mean correlation = $-0.0624 \pm 0.0948$ SD, Table 3). Together, these additional analyses indicate that the correlations obtained are not simply artifacts of our analysis pipeline, but instead reflect an intrinsic relationship between frequency and amplitude well

described by the shape and peak of each subject's alpha bump (Supplementary Figure 3).

**Generalizing the link between amplitude and frequency**. To assess the generalizability of the predictive relationship between frequency and amplitude, we computed PaA for a previously published dataset in which 14 subjects completed four sessions and two subjects completed six sessions of a two-interval contrast discrimination task (with 1176 trials per session; referred to as Experiment 2, for more details see reference 60). In brief, after an attentional cue, two oriented stimuli were presented for 300 ms to the left and right of fixation. After this, there was a blank interval of 600–800 ms followed by a second presentation of two oriented stimuli for another 300 ms. The oriented stimuli were rendered at a variable contrast level ranging from 0% to 81.13% and subjects had to indicate which of the two stimulus presentation intervals contained a slight contrast increment. We focused our analysis on data from the 'divided attention' cue condition in which either stimuli could be the target, because this condition most closely matched the spatial uncertainty of the stimuli in Experiment 1.

Consistent with the first experiment, we observed event-related shifts in average instantaneous frequency and amplitude in the same contralateral and ipsilateral groups of electrodes reported in the first experiment (Fig. 7a). Importantly, these modulations in instantaneous frequency and amplitude are linked, as indicated by high single timepoint correlations between PaA and amplitude ($0.458 \pm 0.063$ SD, Fig. 7b), and the similarity in average PaA and amplitude waveforms (Fig. 7a).

In the more complex paradigm used in Experiment 2, stimuli were presented for 300 ms and were mostly suprathreshold. Thus, unlike Experiment 1, the design of Experiment 2 was not ideal to investigate the impact of alpha modulations on behavioral performance. However, for completeness, we examined the link between alpha amplitude, alpha frequency and behavioral performance using the data from Experiment 2. Like the modulations reported in Experiment 1, we observed lower post-stimulus alpha amplitude in contralateral posterior channels on correct compared with incorrect trials, reflected in an interaction between topography and accuracy ($F(1,14) > 4.44$, Supplementary Figure 5; Supplementary Table 3). In addition, contralateral instantaneous alpha frequency increased before the onset of the first stimulus on correct compared to incorrect trials, but only when stimulus contrast was low (reflected in an interaction between accuracy and contrast with $F(1,14) > 3.31$; Supplementary Figure 5; Supplementary Table 4). The observation of a significant effect only with low-contrast stimuli is consistent with the findings from Experiment 1 in which there was a high degree of sensory uncertainty because stimulus location was unpredictable and the stimuli were low-contrast and masked (see Methods).

In summary, we show that alpha amplitude can be predicted from instantaneous frequency in two different tasks with different

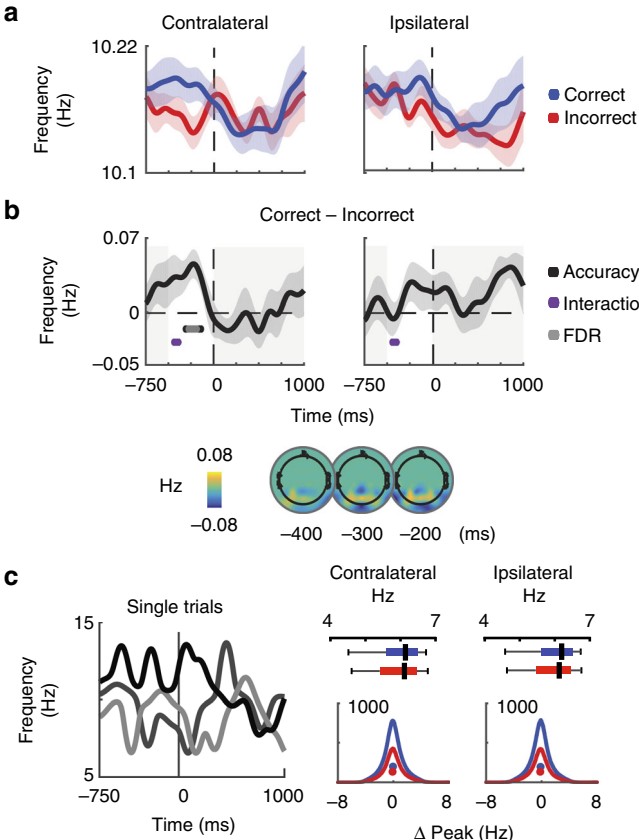

**Fig. 4** Topographically selective increases in pre-stimulus frequency predict accuracy. **a** Contralateral and Ipsilateral electrodes show distinct target-locked patterns in instantaneous frequency. Blue indicates correct trials, red indicates incorrect trials, shaded areas indicate ±1 SEM within subject. **b** A pre-stimulus elevation in frequency on correct as compared to incorrect trials is localized to Contralateral electrodes. All dots indicate significance from zero, evaluated by comparing the obtained t-value with a null distribution of t-values computed by shuffling the condition labels 10,000 times. This analysis was done on a timepoint-by-timepoint basis from −500 ms to stimulus onset, as indicated by the non-shaded areas (see Methods). Significant main effects are indicated in black, whereas gray dots indicate significance after FDR correction at P = 0.05. For illustration, Correct—Incorrect topographies reveal elevated pre-stimulus alpha frequency in 100 ms bins centered around −400, −300 and −200 ms before the stimulus. **c** Three example trials of instantaneous frequency highlight single trial dynamics. Boxplots on the upper right indicate the average single trial dynamic range (max—min) of instantaneous frequency on correct (blue) and incorrect (red) trials. Histograms in the lower right show distributions of instantaneous frequency as a function of the distance from peak alpha over all subjects, timepoints, and electrodes in each of the four conditions. Dots in histograms indicate the median shift for that condition

stimuli and cognitive demands. This suggests that each subjects' amplitude spectrum is a general link between the modulations in frequency and amplitude that correlate with changes in visual perception.

## Discussion

In the present study, we show that alpha amplitude and instantaneous frequency are linked by the spectral characteristics of each subject's alpha oscillation. This result suggests that amplitude and frequency do not reflect unique properties of cortical oscillations. Instead, amplitude may depend on how close

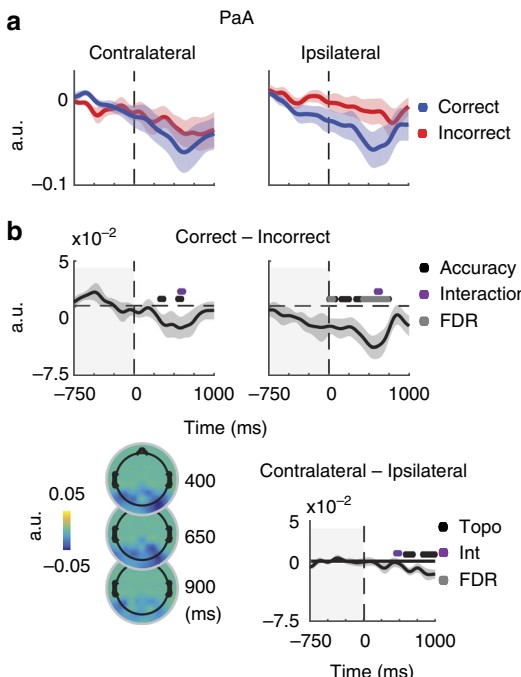

**Fig. 5** Shifts in instantaneous frequency predict alpha amplitude. **a** On a trial-by-trial and timepoint-by-timepoint basis, instantaneous frequency was used to generate predicted alpha amplitudes (PaA). PaA was baselined to the same interval used for alpha amplitude (−1000:−750 pre-target). Shaded areas indicate ±1 SEM within subjects for correct (blue) and incorrect (red) trials. Reported results are averaged over the same groups of contralateral and Ipsilateral electrodes previously reported. **b** Correct—Incorrect differences are plotted for Contralateral and Ipsilateral electrodes. For illustration, topoplots indicate Correct—Incorrect topographies averaged over 100 ms bins centered on 400, 650, and 900 s after stimulus onset. All dots indicate significance from zero, evaluated by comparing the obtained t-value with a null distribution of t-values computed by shuffling the condition labels 10,000 times. This analysis was done on a timepoint-by-timepoint basis from stimulus onset to +1000 ms, as indicated by the non-shaded areas (see Methods). Main effects of accuracy indicated in blue and yellow in contralateral and ipsilateral electrodes, red indicates a main effect of topography. Gray dots indicate significance after FDR correction at P < 0.05

instantaneous frequency is to peak alpha, as predicted by a simple model based on coupled oscillators. Furthermore, modulations of alpha oscillations impact visual information processing in a manner consistent with seperate lines of research that highlight the importance of either alpha amplitude or shifts in instantaneous alpha frequency. We found a contralateral decrease in alpha amplitude and an increase in instantaneous frequency when subjects correctly discriminated a brief target. Historically, these results have been discussed largely in the context of different theoretical frameworks, with amplitude primarily associated with desynchronization[17] and frequency associated with changes in the sampling rate of incoming visual information[35]. However, our results suggest a revision of these traditional accounts and highlight the need for a more unified framework.

In our data, post-stimulus drops in alpha amplitude on correct trials correspond to shifts in instantaneous frequency both above and below peak alpha, as no mean post-stimulus differences in instantaneous frequency are observed between correct and incorrect trials. However, the fact that correlations between frequency and amplitude remain stable after the stimulus suggests that

**Table 2 Amplitude, Frequency and PaA are modulated by experimental conditions**

|  | Amplitude | Instantaneous frequency | PaA |
|---|---|---|---|
| Contralateral vs ipsilateral | *$t(15) = -3.576$, $p = 0.0$ | $t(15) = -1.823$, $p = 0.0827$ | $t(15) = -3.167$, $p = 0.0009$ |
| Accuracy (contralateral electrodes) | *$t(15) = -2.9994$, $p = 0.0006$ | *$t(15) = 3.399$, $p = 0.0015$ | $t(15) = -2.279$, $p = 0.0141$ |
| Accuracy (ipsilateral electrodes) | *$t(15) = -3.878$, $p = 0.0$ | $t(15) = 1.583$, $p = 0.135$ | *$t(15) = -3.597$, $p = 0.0001$ |
| Location×accuracy interaction | *$t(15) = -3.197$, $p = 0.0002$ | $t(15) = 2.549$, $p = 0.0248$ | $t(15) = 2.134$, $p = 0.0399$ |

The empirically observed amplitude, instantaneous frequency and predicted alpha amplitude (PaA) as a function of electrode location and behavioral performance. All tests report the maximum or minimum timepoint-by-timepoint $t$-values over a temporal window extending from target onset to 1000 ms after target onset for amplitude and PaA, and from −500 ms to target onset for frequency. $t$-values were compared against distributions obtained empirically by randomizing condition labels 10,000 times and then repeating the same statistical test (see Methods). Reported $t$-values are from the timepoint with smallest $p$-value. * indicates that $p$-values were significant after FDR correction at alpha = 0.05 from stimulus onset to + 1000 ms (amplitude and PaA) or −500 ms to target onset (instantaneous frequency)

**Table 3 Correlations are specific to the alpha bump**

| Correlation (mean $\pm$ SD) | PaA$_{Noise}$ | PaA$_{Shuffled}$ | PaA$_{1/f}$ | PaA$_{Empirical}$ |
|---|---|---|---|---|
| Correct | $-3.35*10^{-20} \pm 9.2*10^{-16}$ | $-0.0173 \pm 0.1209$ | $-0.0624 \pm 0.0948$ | *$0.4745 \pm 0.0728$, $p = 0$ |
| Incorrect | $4.08*10^{-20} \pm 6.8*10^{-16}$ | $-0.0183 \pm 0.1318$ | $-0.0594 \pm 0.1045$ | *$0.4802 \pm 0.0736$, $p = 0$ |

Control look-up table analyses were performed to generate PaA$_{noise}$, PaA$_{shuffled}$, and PaA$_{1/f}$, which were then correlated with amplitude (see Methods). Average correlation coefficients $\pm$ standard deviations are shown for all analyses. PaA$_{noise}$ was generated with a series of white noise spectra as look-up-tables, producing small correlations with amplitude indistinguishable from 0. PaA$_{shuffled}$ was generated by repeatedly shuffling the frequency axis of a given look-up-table, but again PaA$_{shuffled}$ was uncorrelated amplitude. PaA$_{1/f}$ was generated with look-up-tables captured the 1/f component but did not contain the characteristic alpha bump. PaA$_{1/f}$ was also uncorrelated to the empirically observed alpha amplitudes. *indicates significance of empirically obtained PaA values, computed by comparing $t$-tests against zero of these values to $t$-tests the shuffled PaA values, and then FDR correcting at $P = 0.05$

that changes in instantaneous frequency are related to those in amplitude. To understand this, it is important to remember that amplitude and frequency do not have a unique, one-to-one mapping. Instead, they are related by the non-monotonic bump shape of the amplitude spectra. This means that significant differences in one metric may average out in another metric. For example, amplitude could be equal at timepoints in which instantaneous frequency has moved from below to above peak alpha (or vice versa). Thus, it is possible that before the stimulus, an increase in instantaneous frequency enhances perception, but upon stimulus presentation a shift either above or below peak alpha enables efficient visual information processing.

These observations suggest a possible mechanism for how the desynchronization of alpha oscillations results in efficient sensory processing. In the desynchronization account, fewer visual neurons are entrained at a common alpha frequency as activity in relevant circuits shifts to process sensory stimuli. The current data suggest that shifts in instantaneous alpha frequency predict changes in alpha amplitude, which could be interpreted as a mechanism for this "drop out". Shifts in instantaneous alpha frequency away from the peak or resonant frequency—akin to a pianist drifting from a metronome—may be the mechanism by which desynchronization and drops in alpha amplitude occur. At a neural level, these frequency changes could occur when the E/I balance shifts to allow the formation of local circuits that process relevant sensory stimuli[16,67–69]. For example, changes in the activity of specific sub-sets of inhibitory interneurons likely modulate the instantaneous frequency of the local circuit[14,49,70–72]. Thus, increases and decreases in instantaneous frequency could be due to a variety of changes in the E/I balance during sensory processing, and future research will be required to determine the contribution of factors such as dampening, changes in a region's characteristic frequency, and changes in the driving region's characteristic frequency.

Finally, in addition to alpha amplitude and frequency, several previous studies have found a correlation between behavioral performance and alpha phase[24–26]. Although we did not find consistent dependence of performance on alpha phase, the non-stationarities that we observed in instantaneous frequency might impair our ability to detect performance-related phase offsets[41,66]. Further work is needed to understand how frequency shifts might contribute previously reports of phasic modulations in perceptual sensitivity.

In sum, our results show that fluctuations in the instantaneous frequency of alpha oscillations are associated with both behavioral performance and alpha amplitude. This suggests that changes in instantaneous frequency and amplitude do not reflect completely independent mechanisms for mediating visual information processing, and our results provide new insights into understanding how coupled changes in oscillatory frequency and amplitude jointly impact visual information processing.

## Methods

**Subjects.** In Experiment 1, 17 subjects (eight male) were recruited at the University of California San Diego and all data were collected at UCSD's Perception and Cognition Lab. The age range of the subjects was 19–30 years old (22.06 mean $\pm$ 3.98), and all subjects had normal or corrected to normal vision. All subjects provided written informed consent in accordance with the Institutional Review Board at UCSD. Subjects were compensated $10/h for behavioral training and $15/hour for EEG. One subject was excluded owing to a high number of independent components showing blink related activity (i.e., five frontally localized components exceeding > 30 mV).

Experiment 2 is described in detail in Itthipuripat et al 2014[60]. In brief, 17 subjects (18–31 years old, nine females) underwent a 2.5-h behavioral training session and then 14 subjects completed four EEG sessions and two subjects completed six EEG sessions for a total of 4704 or 7056 trials, respectively. One subject withdrew after the second EEG session, yielding 16 subjects for the final analysis.

**Apparatus and stimuli.** The experiment was implemented using Psychtoolbox in the MATLAB programming environment running on a Windows PC with the XP operating system. Subjects were positioned 60 cm from the display and stimuli were presented on a 15-inch CRT monitor with $1024 \times 768$ resolution and 120 Hz refresh rate. The luminance output of the monitor was measured using a Minolta LS110 and linearized in the stimulus presentation software.

In Experiment 1, all stimuli appeared 8.5° of visual angle to the left or to the right (with equal probability) of the central fixation point (with 0° offset from the horizontal meridian). At the start of each stimulus presentation sequence, a disk of Gaussian white noise (5.7° diameter) was presented for one video frame (8.33 ms) in one of the two possible locations. Next, either a vertically or horizontally oriented Gabor target stimulus was presented for one video frame in the same spatial position as the white noise stimulus (also 5.7° diameter). Following the offset of the Gabor, a second white noise stimulus was presented for one video frame. Subjects reported whether the orientation of the Gabor stimulus was vertical

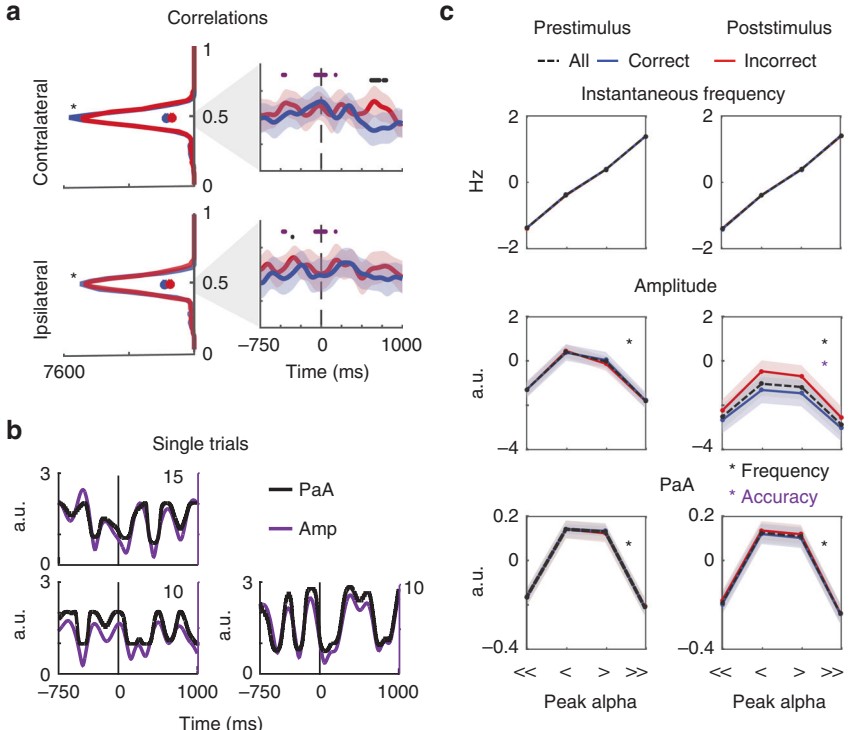

**Fig. 6** Predicted alpha amplitude correlates with observed amplitude. **a** To assess how well PaA corresponds with alpha amplitude, we computed correlation values for each subject and electrode for each timepoint over the entire –750 to 1000 ms peri-stimulus interval. Histograms show correlation values concatenated over all subjects, timepoints and electrodes on correct (blue) and incorrect (red) trials in the contralateral and ipsilateral electrodes. Stars on each panel indicate that all correlation values shown in these histograms are significantly different from correlations obtained with shuffled LUTs (see Methods, Supp Fig. 4). Dots in histograms indicate the median correlations for that condition. Timecourses panels on the right show these correlations are relatively stable over time, where the y-axes of the plots run from 0.425 to 0.525, corresponding to the gray shaded area in the histogram. All dots indicate significant difference in the correlations between conditions, evaluated by comparing the obtained t-value with a null distribution of t-values computed by shuffling the condition labels 10,000 times. This analysis was done on a timepoint-by-timepoint basis from −500 to +1000 ms, as indicated by the non-shaded areas. Main effects of are in black, whereas purple indicates an interaction. Gray dots indicate significance after FDR correction at 0.05. **b** Three example trials from three different subjects show that PaA shifts on single trials mirror those in alpha amplitude. The y-axis and traces for PaA are indicated in black, while those for amplitude are purple. Note that the y-axis range is different in each subplot to maximize visibility of amplitude and PaA (see Methods). **c** Trials were sorted according to mean pre (−350:−50 ms) and post (350:650 ms) stimulus frequency. Average amplitude and PaA were then computed on these trials and timepoints. Trials were further split by accuracy, as indicated by blue and red lines. Significant differences were evaluated using a two-way repeated-measures ANOVA. Black stars indicate a significant effect of frequency, whereas a purple star indicates a significant effect of accuracy in the two-way ANOVA

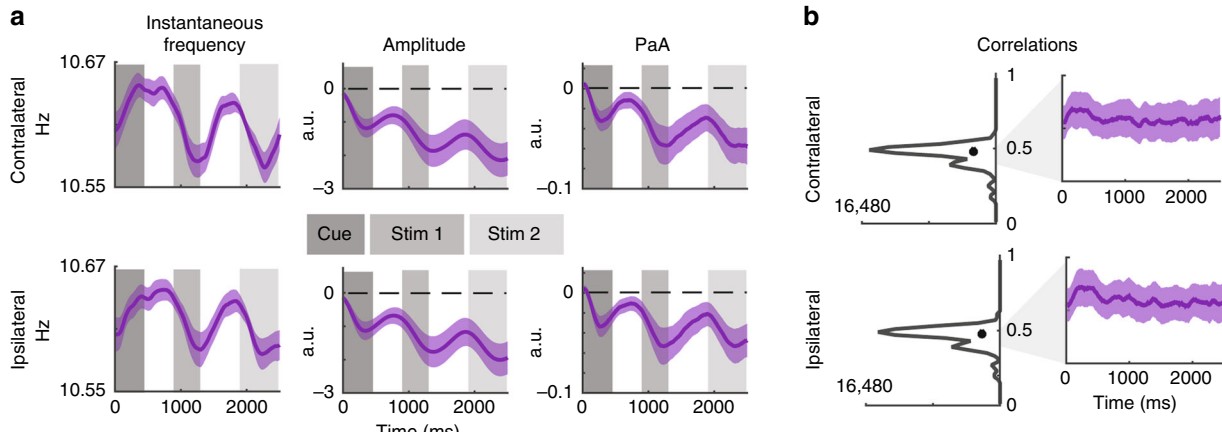

**Fig. 7** Frequency, amplitude and PaA in Experiment 2. **a** Average instantaneous frequency, amplitude, and PaA in the same contralateral and ipsilateral electrodes examined previously, shaded areas indicate ±1 SEM within subjects. All data are locked to the onset of the cue (indicated by dark shading). Alpha amplitude shows event-related decreases corresponding to the onset of the cue, stimulus array 1 and stimulus array 2. Similarly, the rightmost panel shows that average shifts in PaA mirror these changes in amplitude. **b** Histogram of single trial correlations collapsed across subjects, timepoints, and electrodes. Traces to the right indicate timecourses of these correlations. Timecourses show these correlations are relatively stable over time, where the y-axes of the plots run from 0.4 to 0.5, corresponding to the gray shaded area in the histogram

or horizontal by pressing one of two buttons on a small keypad. Subjects were instructed to respond as quickly as possible, and to do their best to avoid blinking until after a response was made. After subjects responded, there was an inter-target-interval of 3000–4000 ms (pseudo-randomly sampled from a uniform distribution). Each experimental block (72 trials) lasted for ~7 mins. Subjects completed 14 blocks of trials during the EEG recording session.

The main goal of Experiment 1 was to determine whether frequency shifts in the alpha band predicted behavioral performance. Before the EEG recording session, the contrast threshold yielding vertical/horizontal discrimination accuracy between 60 and 65% was determined in a separate thresholding session using the method of constant stimuli. In the EEG recording session, the mean accuracy across subjects after trial exclusion was 65% ± 2.8%, and mean contrast was 5.02% ± 1.12% (mean ± SD). Aside from titrating contrast to estimate the threshold for each subject, the stimulus presentation sequence and timing of the trials in the behavioral and the EEG sessions were identical.

In Experiment 2, subjects performed a two-interval forced choice contrast discrimination task in which each trial began with a 500 ms cue instructing subjects to attend to locations in either the left, right or both hemifields (100% valid)[60]. The cue was followed by a 400–600 ms inter-stimulus interval in which only the fixation point was present. At a pseudo-randomly chosen time within this inter-stimulus interval window, a first stimulus pair was presented (two sinusoidal Gabor patches, one in each hemifield) for 300 ms, where each Gabor was presented at one of seven pedestal contrasts. After another 600–800 ms inter-stimulus interval in which only the fixation point was visible, a second pair of Gabors was presented for 300 ms. A small contrast increment was added to the pedestal contrast of the target Gabor patch during either the first or second presentation interval, and subjects were asked to report if the increment occurred during the first or second presentation. For the first six pedestal levels, the magnitude of the contrast increment was adjusted to maintain ~76% accuracy, while accuracy the highest pedestal contrast level could not be titrated because the contrast was too high and so was not included in the analysis in Supplementary Figure 5 (consistent with exclusion of that condition in the published manuscript[60]). In addition, several aspects of this design make it conceptually different from the relatively simple design employed in Experiment 1 to examine frequency and amplitude modulations. These include the longer (300 ms) stimulus presentation, the reliance of the task on working memory during the delay interval, the presentation of bilateral stimuli, and the use of different pedestal contrasts in each hemifield on each trial (as contrast is known to modulate frequency[66]).

**EEG recording**. All EEG recordings took place in a sound-attenuated and electromagnetically shielded room (ETS Lindgren, Cedar Park, TX, USA). EEG and electrooculogram were recorded with a Biosemi Active2 System (Amsterdam, The Netherlands) using a headcap with standard Biosemi 64 electrode layout. In addition to the 64 scalp electrodes, one reference electrode was placed on each mastoid, and 6 electrodes were placed around the eyes to identify and reject trials with blink and saccade artifacts. All EEG data were recorded at a sampling rate of 512 Hz. Event triggers were recorded in the EEG data file to mark the time of target presentation and the time of the subject's response.

**EEG preprocessing**. After data collection, data from the scalp electrodes were re-referenced to the algebraic mean of the two mastoid electrodes. Then, the raw timeseries from each electrode was bandpass filtered between 0.1 and 55 Hz using a third-order Butterworth filter to attenuate slow drift and 60 Hz line noise. After filtering, data were epoched into 6-second intervals centered on the presentation of each target. Trials were excluded from further analysis if the electrooculogram electrodes located above or below either eye reached ± 85 mV (blinks) or electrooculogram electrodes located outside either outer canthi reached ± 45 mV (saccades) within ± 1 second of target presentation (7.8% ± 8% S.D. of trials were excluded). Additionally entire blocks of trials were rejected when there was a failure to record the precise timing of any of the target onsets (i.e., a trigger that was sent to the EEG recording software was not recorded: 5 out of 256 total blocks across all subjects). For each subject, electrodes showing voltage fluctuations exceeding the 95th percentile of data from all electrodes and timepoints were also excluded from further analysis (1.8 ± 1.8 S.D. electrodes excluded). Finally, trials with RTs > 2000 ms were excluded from further analysis (another 4.8% of trials). After applying these exclusion criteria, subjects had an average of 868 ± 89 SD trials, 35% of which were incorrect. Thus, a proportionate number of correct and incorrect trials were rejected owing to artifacts. In addition, after artifact rejection, 50.25% (range 48.7–53.2% across subjects) of remaining target presentations were on the left side of the screen, indicating that artifacts were distributed equally between left and right targets. Performance was quite stable across the course of the EEG recording session (paired t-test comparing accuracy in the first and last block t(15) < 0.058, p > 0.95).

**Statistics**. For all analyses, we report results from contiguous groups of 3 electrodes of interest (EOIs) located over the left and right occipital cortex identified *a priori* based on previous studies—namely: P3, Po7, and Po3 over left occipital cortex and P4, Po8, and Po4 over right occipital cortex[27,55,56,60]. All data are arranged according to target location such that electrodes were subsequently referred to as contralateral and ipsilateral electrodes throughout the paper. Finally,

all statistical comparisons were paired *t*-tests where *p*-values were computed using an empirical null distribution of *t*-values computed by randomizing condition labels 10,000 times (except for Fig. 6c and Supplementary Figure 5 where analysis of variances were used, see below). For example, to compare responses between contralateral and ipsilateral electrodes, we generated an empirical null distribution by pseudo-randomly swapping or maintaining the contralateral/ipsilateral labels on each trial for each subject and then repeating the entire statistical analysis pipeline as normal (and this procedure was repeated 10,000 times). Thus, note than any *p*-values reported as 0 indicate that the observed effect was larger than any of the 10,000 iterations of this randomization procedure. For consistency across analyses, we also used *t*-tests to difference scores to evaluate interaction terms, in which case the *t*-values we report are equivalent to the square root of the *F*-values that are produced by an analysis of variance. For ERPs, statistical comparisons were performed on average amplitudes in 50 ms time windows centered on peak latencies in the grand average waveforms[73], or from 210–260 and 460–510 ms post stimulus for the END and LPD respectively[25,52–54,60]. Note that our slightly delayed ERP epochs (when compared with some previous studies) are consistent with the low-contrast of our stimulus and difficulty of our task[25,54,62]. Otherwise, statistical comparisons were performed at each sample in either a 500 ms pre-stimulus epoch (for instantaneous frequency) or a 1000 ms post-stimulus epoch (amplitude, PaA), based on previous studies[17–20,22,24–26]. Statistical comparisons of the correlations between real alpha amplitude and PaA were performed over the entire 1750 ms epoch to err on the side of being conservative as there is no precedent in the literature. All p-values were then FDR corrected at $p <= 0.05$[74].

For the analysis in Fig. 6c, trials were sorted based on their average frequency in either a pre (−350:−50 ms) or post (350:650 ms) stimulus epoch based on timepoints significant for Figs. 3b and 4c. Specifically, the lowest («) bin consisted of trials in the lower half of a median split of trials with a mean frequency below peak alpha. Accordingly, the second lowest ( < ) bin were trials in the upper half of a median split of trials with frequency below peak alpha. The > and » bins were computed similarly, but were composed of trials with means greater than peak alpha. Average amplitude and PaA were then computed for these trials and epochs. A two-way repeated-measures analysis of variance with frequency bin and accuracy was used to assess how amplitude and PaA depended on frequency in these epochs, and *p*-values were computed by comparing observed *F*-values to a distribution obtained from 10,000 randomizations of condition labels. Finally, Supplementary Figure 5 and Supplementary Tables 1 and 2 were computed using a three-way repeated-measures analysis of variance with the pedestal contrast of the target (collapsed across consecutive pedestals to yield three instead of six levels), accuracy and topography as factors. The analyses in Supplementary Figure 5 and Supplementary Table 3 and 4 use only the divided attention trials to make the interpretation of these timecourses more comparable to those analyzed in Experiment 1 (in which the location of the target was not pre-cued). *p*-values were computed by comparing *F*-values to distributions obtained by shuffling condition labels 5000 times.

**ERPs**. ERPs were obtained by averaging stimulus-locked timecourses for each electrode of interest and then using a low-pass third-order Butterworth filter with a cutoff frequency = 5 Hz. All time-frequency analyses were performed using custom MATLAB scripts (see below for details). To avoid edge artifacts, all filtering was applied to 6 s epochs centered on stimulus presentation, after which peri-stimulus time epochs of interest were extracted (i.e., epochs ± 1,000 ms around the presentation of each target). Note that all statistical analyses were performed on data before the 5 Hz low-pass filter was applied. The low-passed data were presented in the figures for visualization purposes only. Also note that there was not a pronounced P1 component (assessed by using a cutoff frequency of 15 Hz), consistent with the use of a low-contrast or briefly presented target stimulus[25,60].

**Alpha amplitude**. The timecourse of stimulus-locked alpha amplitude at each electrode's peak alpha frequency was extracted by bandpass filtering the data with a third-order Butterworth filter spanning ± 2.5 Hz centered on the peak frequency to EEG data from each electrode and subject and then applying a Hilbert transform to this filtered timeseries. As in the preprocessing of the EEG data for generating ERPs (see above), we applied the bandpass filter to a 6000 ms epoch surrounding target onset to avoid contaminating the peri-stimulus window (±1000 ms) with edge artifacts. Alpha amplitude on trial *k* at time *t* was estimated by Hilbert transforming the bandpassed timeseries to yield a complex representation of the form $Ce^{i\omega}$. Note that C describes the amplitude and $\omega$ the frequency of the signal. Thus, we take the absolute value of these complex coefficients to yield an amplitude estimate:

$$A_{k(t)} = \left| C_{k(t)} e^{i\omega_{k(t)}} \right|$$

All amplitude values were then baselined on a trial-by-trial basis by subtracting the mean amplitude −1000 to −750 ms before the stimulus.

**Instantaneous frequency**. Instantaneous frequency is defined as the first derivative in time of the phase of the EEG signal, or the change in phase per unit time as time approaches zero (see ref. [66] for review). For each subject and EOI, artifact free epochs were bandpass filtered at ± 2.5 Hz around peak alpha using a 3rd order Butterworth filter (again, bandpass filtering done on 6000 ms epochs surrounding

target onset to attenuate edge artifacts in the peri-stimulus window). We then applied a Hilbert transform to the filtered data from each epoch to obtain the amplitude and phase of the EEG response at each point in time on each trial. The phase angle was unwrapped to be cumulative so that there were no discontinuities at −pi and pi. We then calculated instantaneous frequency by approximating the derivative of these unwrapped phase angles. To yield an estimate of frequency in Hz at time $t$ and trial $k$, we then normalized this approximate derivative by the sampling rate ($sr$). Because computing numerical derivatives of discretely sampled timeseries can produce sharp discontinuities, we attenuate the influence of these outliers by low-pass filtering our estimates of the derivative of the phase angle. More formally, we estimated the instantaneous frequency on trial $k$ and time $t$ by fitting a line of the following form to the unwrapped phase data in temporal window of 88 data samples centered on time $t$: .

$$p_{t,k} = d\phi_{k(t)}\mathbf{x} + I_{t,k}$$

Where $p_{t,k}$ corresponds to the estimated unwrapped phase, parameterized by scalar $d\phi_{k(t)}$, or an estimated slope (change in phase angle $\phi$), vector $x$, the time axis, and scalar $I_{t,k}$, the y intercept. The window size of 172 ms for $x$, corresponding to 88 data samples at $sr = 512$ Hz, was selected because it was the smallest window that kept the average instantaneous frequency fluctuations on single trials within the 5 Hz wide bandpass range (Supplementary Figure 1). Given this fit, we defined instantaneous frequency at time $t$ and trial $k$:

$$\omega_{inst}(k, t) = \frac{d\phi_{k(t)}}{2\pi} * sr + I_{t,k}$$

Where $\omega_{inst}(k, t)$ corresponds to an estimate of the instantaneous frequency at time $t$ on trial $k$. The regression lines were estimated using a least squares fitting algorithm to the unwrapped phase data and the fits were generally quite good ($R^2 = 0.995 \pm 0.016$, mean $\pm$ SD). We also evaluated our results by estimating instantaneous frequency by simply subtracting sequential points along the timeourse of the unwrapped phase:

$$d\phi_{k(t)} = \phi_{k(t+1)} - \phi_{k(t)}$$

$$\omega_{inst}(k, t) = \frac{d\phi_{k(t)}}{2\pi} * sr$$

However, owing to occasional sharp discontinuities in the first derivative, this second method then requires the application of median filters over large temporal windows to attenuate the influence of fluctuations far outside of the bandpass range (see[35,66]). In our data sets, both methods yielded similar results.

**Look-up-tables relating frequency and amplitude**. To generate the look-up-tables (look-up-tables) that were used to relate changes in instantaneous frequency and amplitude, we used a wavelet decomposition based on a family of Morlet functions with center frequencies ranging from 3 to 20 Hz in 0.1 Hz steps. Using these wavelets, the amplitude at each frequency in this band was estimated and stored for use in the main analysis. To avoid biasing the results, the amplitude look-up-tables were calculated from a set of 6-s-long epochs drawn from an equal number of correct and incorrect trials separately for each subject and electrode (mean number of trials across subjects: $325 \pm 36$ SD). These spectra were also used to define peak alpha for each subject and EOI ($10.3$ Hz $\pm 1.1$ Hz SD across subjects in the 6 posterior electrodes of interest). The use of averaging many long epochs to estimate amplitude spectra (for our look-up-tables) is related to Welch's method[75]. This method is common in spectral density estimation for achieving both (1) high frequency resolution and (2) low variance and stability in the estimate. Thus, this procedure produces stable amplitude spectra for each subject and electrode. In fact, within a subject the three contralateral channels we analyzed are correlated at 0.97 $\pm 0.02$ SD, illustrating that our technique tends to converge on similar, stable spectra for neighboring channels. In contrast, the three contralateral channels show a much weaker correlation of $0.69 \pm 0.09$ SD between subjects, confirming that these spectra are phenotypic and subject specific.

To generate white noise look-up-tables, we used the built in white Gaussian noise (wgn) function in Matlab with parameter output power set to 1 dBw. We generated 6 s epochs of white noise separately for each subject using the number of trials in their dataset. Look-up-tables were then estimated by using a wavelet decomposition of these trials as described above. This process was repeated for 5000 iterations so we could assess the stability of resultant PaA estimates.

Shuffled look-up-tables used for statistical comparison were computed by pseudo-randomly shuffling the frequency axis (3 to 20 Hz in steps of 0.1) of each subject and electrode's original look-up-table 5000 times.

Finally, to generate 1/f look-up-tables, we fit a two-term exponential to each subject and electrode's original look-up-table using Matlab's built in fit function and excluded the alpha bump (i.e., fit only amplitudes at frequencies below 5 and above 14 hz) in the fitting procedure.

**PaA**. We evaluated the hypothesis that changes in alpha amplitude and shifts in instantaneous frequency are interdependent by using each subject's amplitude spectra as a look-up-table to link these two metrics (as shown in Fig. 1b). On every

trial and timepoint, instantaneous frequency was used to index into this look-up-table, yielding a predicted amplitude value for each timepoint and trial. The alpha amplitude distribution is known to be a stable trait[76]—hence we are using each subject's phenotypic amplitude spectrum to generate single trial PaA (Supplementary Figure 2b). Single trial examples of PaA and amplitude are shown for three subjects (2, 13, and 15) on trials and electrodes 814, 623, 193 and 30, 26, 63, respectively (Fig. 6b).

**Correlations with observed alpha amplitude**. We correlated PaA timecourses—computed by passing instantaneous frequency through the amplitude look-up-table—with empirically observed amplitude timecourses on a timepoint-by-timepoint basis. Note that these correlations emphasize the similarity of the timecourses as opposed to matching the exact scaling of the PaA with respect to the scale of the empirically observed data. Indeed, differences in the overall magnitude of PaA and the observed amplitude vary because (a) wavelet transforms were used to estimate the look-up-tables while Hilbert transforms on bandpassed data were used to generate empirical estimates of alpha amplitude and (b) stable amplitude look-up-tables result from averaging many trials, and thus PaA reflect these average magnitudes as opposed to the single trial magnitudes for observed amplitude. We used wavelets to generate the look-up-tables so that we could increase the frequency resolution of our look-up-tables (i.e., smaller step sizes along the x-axis in Fig. 1b). In Experiment 1, we computed correlations over 2000 ms epochs centered on target presentation. In Experiment 2, we computed correlations over 4000 ms epochs locked to an attentional cue that occurred 0.5 s into each trial.

**Phase locking and phase bifurcation index**. To make contact with previous papers, we also examined the relationship between alpha phase and behavioral performance. We first computed the intertrial phase locking index (PLI) by applying Hilbert transforms to data bandpassed around peak alpha as described previously. PLI was estimated from the complex values obtained from this Hilbert transform at time $t$ over trials 1 to $k$ using the formula:

$$PLI(t) = \frac{1}{k} \sum_{1}^{k} \frac{C_{k(t)} e^{i\omega_{k(t)}}}{\left| C_{k(t)} e^{i\omega_{k(t)}} \right|}$$

Where $Ce^{i\omega}$ is the same complex representation of the data as outlined in the amplitude section above. This value ranges from 0 to 1 (no phase locking to perfect phase locking at any phase). As stimulus onset was unpredictable, pre-stimulus alpha phase should be randomly distributed over all trials. Thus, we computed a phase bifurcation index from PLI to assess whether any observed phase locking occurred at the same or opposite phases between accuracy conditions (as described in[25,26]). Bifurcation was computed over correct and incorrect trials at time t:

$$B(t) = (PLI(t)_{correct} - PLI(t)_{all}) \times (PLI(t)_{incorrect} - PLI(t)_{all})$$

Note that this value ranges from 1 (perfect phase locking in both conditions at opposite phases, leading to $PLI_{all} = 0$) to −1 (perfect phase locking in only one condition). Values close to zero indicate random phase distributions for correct and incorrect trials (Supplementary Figure 4,[25]).

**Data availability**. EEG data and Matlab code supporting the frequency and PaA findings of this study have been deposited in the open science framework, accessible at https://osf.io/wkx5h/. Further data that support the findings of this study are available from the corresponding author upon reasonable request.

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

## Acknowledgements

We thank Sarah Fraley for help with data collection, and Eran Mukamel, Vy Vo, Bradley Voytek, Tommy Sprague, and Bradley Postle for useful discussions. Supported by NSDEG graduate fellowship to S.N., by HHMI international student fellowship and a Royal Thai Scholarship from the Ministry of Science and Technology, Thailand to S.I., and by NEI R21-EY024733, NEI R01-EY025872, and James S. McDonnell Foundation awards to J.T.S.

## Author Contributions

S.N. and J.S. conceived the idea. S.N. carried out EEG experiment and analysis of data. R. S. provided feedback on time-frequency decomposition methods. S.I. provided data for analysis of relationship in Experiment 2. S.N and J.S. co-wrote the paper. All authors discussed the results and commented on the manuscript.

## Additional information

**Competing interests:** The authors declare no competing financial interests.

