## [Peer Review File · Nature Communications]

Reviewers' comments:

Reviewer #1 (Remarks to the Author):

The authors present a synchronization analysis of EEG signals studying the alpha band to infer mechanisms on behavioral performance. They mainly show that both amplitude and phase are interrelated there and one has to consider both.

I am not a specialist in neuroscience and may be this is an innovative result for that field.

But from the complex systems science aspect (which I know much better) this phenomenon has been well understood and for basic systems it was clearly explained under which conditions a synchronization only among phases is possible and under which an interrelation of phases and amplitudes is necessary for reaching synchronization (cf. the review by Boccaletti et al., Phys. Rep. 2002 and references therein).

Also the techniques for estimating phases and amplitudes used here are well known and were published several years ago (all of that is not cited in this contribution). There are also more advanced techniques on statistical tests for such estimates as applied here.

Therefore, I cannot recommend this contribution for publication in Nat. Comm.

Reviewer #2 (Remarks to the Author):

The study by Nelli et al represents a milestone for a better understanding of brain oscillations and alpha in particular. Particularly interesting - and novel - is the theoretical background as outlined in Fig.1. The prediction that correct trials are associated with an increase in (instantaneous) frequency and decrease in amplitude is well in line with the well documented observation that desynchronization in the upper alpha band is closely associated with the (cognitive) performance of a task. The article is well written, the experimental design and the results are convincing. I also want to say that the cited references represent an up to date review of relevant work. I have no concerns whatsoever.

The only thing that might improve the impact of the paper is a more detailed discussion of the results with respect to the length of the epochs (6 sec) on which the calculation of the spectra (for the LUT's) and the short time periods for the instantaneous and performance related changes in instantaneous frequency and Amplitude changes.

Reviewer #3 (Remarks to the Author):

Neural oscillations measured at the scalp with electroencephalography have been linked with a variety of aspects of perception and cognition. One of the dominant oscillations is in the alpha frequency band (~8-14 Hz). Alpha oscillations have been the focus of a lot of recent work and the submitted manuscript represents a timely contribution because it

reveals an association between amplitude, instantaneous frequency, and behavior. More specifically, the authors show that both amplitude and instantaneous frequency are not only related to behavior, but fluctuations in frequency can predict amplitude on single trials. This contribution is important because a) amplitude and instantaneous frequency are often thought to be independent and b) it helps to link two explanations of the relationship between alpha oscillations and perception. The manuscript is extremely well-written and clear, the analyses are novel and include appropriate controls (especially using individual spectra to make LUTs), and the results are clear. There is no question in my mind that this work is appropriate for Nature Communications

I have only a few minor issues that should be pretty easy to address.

1. Studies that have shown a relationship between prestimulus oscillations and behavior, typically use a cue. Here, however, there is no cue and yet prestimulus increases in instantaneous alpha frequency on correct trials, but only contralateral to the upcoming stimulus. Given that participants couldn't predict where the stimulus would occur and there was only limited temporal predictability, this result seems a bit unexpected. Could the authors provide a bit more explanation? Does this change reflect some volitional process or is it more spontaneous?
2. I understand the point of the analogy on p. 4. Given that it is not perfect, I'm wondering if there are a couple of steps that could be made to tighten it up a bit. For example, the "damping mechanisms" that constrain the amplitude of the jumps could be more explicitly linked with those that dampen neural oscillations.
3. In the methods, the authors are pretty clear that they use a Butterworth bandpass when computing alpha amplitude. However, they also indicate that the data are also bandpassed during initial preprocessing, but it is unclear what was used for this bandpass.

Signed

Barry Giesbrecht

Reviewer #4 (Remarks to the Author):

In this paper, the authors attempted to closely examine the relationship between the instantaneous frequency and its amplitude of occipital alpha power. Although previous studies have highlighted the importance of each oscillatory signal in visual cognition, there hasn't been a systematic effort to link the two in a mechanistic manner. In that respect as well as the surging interest in utilizing and understanding EEG oscillatory signals as temporarily resolved signals of cognitive operations, I believe that such attempt would be of great impact and interest to the field. This being said, however, although I have no doubt this paper made an important first step in suggesting a case for the interdependence of the two oscillatory signals, I don't think that the current version of the manuscript is strong enough to warrant a publication at this journal. I would also like to emphasize that my concern can very well be resolved by additional analyses as well as re-phrasing of the manuscript.

The key finding in this manuscript is that by knowing the instantaneous frequency at a given moment, we can reliably predict the instantaneous amplitude of the occipital alpha power.

This clearly demonstrates the interdependence of the instantaneous alpha amplitude on the instantaneous frequency, but when you examine the time course of instantaneous frequency and amplitude on correct v.s. incorrect trials, they do not exhibit the same time course. More precisely, the instantaneous alpha amplitude decreases after stimulus onset in a bilateral fashion with stronger suppression on the contralateral side to the stimulus. However, the effect of instantaneous frequency on correct v.s. incorrect trials is limited to a couple hundred milliseconds prior to the stimulus onset. Given these time course differences, it is very interesting and at the same time puzzling as to why the PaA generated by the look-up table analysis resulted in the pattern resembling the actual amplitude data. With regard to this puzzle, the authors do suggest an interesting hypothesis. The pre-trial shift in instantaneous alpha frequency represents the shift in the "sampling rate" of our visual system while post-stimulus shift in alpha amplitude (likely induced by both increase and decrease of alpha frequency, or pure decrease in amplitude unaccompanied by a frequency shift) represents efficient visual information processing. To potentially better answer this puzzle, I would like to see whether there is trial-type dependent differences in the post-stimulus shift in instantaneous frequency. One potentially interesting approach might be to classify trials based on the post-trial history of frequency shift and examine both amplitude shift as well as behavioral output. These analyses can potentially clarify the seemingly puzzling findings and provide additional insight into their interpretation of what instantaneous frequency and amplitude modulation reflect in terms of visual cognition. Another concern that I had was the generalizability of the findings. The authors did report the similar interdependence of the instantaneous frequency and the amplitude modulation in two different experiments, but their analyses for Experiment 2 seemed incomplete. For instance, what was the nature of the time course of instantaneous frequency effect and amplitude effect? Were they more or less consistent with what they found in Experiment 1 (i.e. pre-stimulus increase in contralateral instantaneous frequency for correct trials and post-stimulus bilateral amplitude suppression (larger for contralateral channels) for correct trials)? The readers would benefit a lot from knowing these details in judging not only the consistency of the findings as well as the validity of the authors' interpretation as to what cognitive mechanisms the instantaneous frequency and amplitude effect might reflect. Taken together, although I have no doubt that this paper will make a great impact to the field, I think that the current manuscript would benefit a lot from a major revision.

Minor points.

The baseline window for the amplitude analyses was set to -1000 to -750ms before the stimulus onset. Was there a specific reason why this window was chosen?

Figure 4.

Although I was able to figure out the color-coding for panel A thanks to its consistency with other figures, it is a good idea to put a legend on every panel.

Page 14

The first sentence of Instantaneous Frequency section.

Please capitalize the first "i" in the "instantaneous".

Thank you for your time and for the constructive comments regarding our manuscript. Below we provide detailed replies to each of the reviewer's comments (our comments in blue). We addressed all the concerns/questions that were raised and we would be happy to address any remaining issues. We begin with a list of general changes to the manuscript before moving on to specific replies to each of the reviewer's comments.

General comments to all reviewers:

Our primary focus in this manuscript is on the interdependence between frequency and amplitude as opposed to how either frequency or amplitude are independently related to behavior. However, to make better contact with the existing literature, the revised manuscript further explores how frequency and amplitude differ as a function of behavioral performance. We added a new panel E to Figure 5 to intuitively illustrate how amplitude and frequency vary within the pre- and post-stimulus epochs that showed significant modulations in Experiment 1. With respect to Experiment 2: the primary purpose of including this previously published data set was to provide an independent evaluation of the interdependence between amplitude and frequency in an experimental paradigm substantially different from Experiment 1. Nonetheless, in the revision, we now include a more thorough analysis of these data. While we see some common trends between the alpha modulations and behavior across the two experiments, as the data from Experiment 2 was chosen for its dissimilarity to Experiment 1, there are several reasons why discrepancies are to be expected as well (which we now discuss more thoroughly in the Results section of the manuscript). The key result, however, is that in both experiments there is a strong relationship between alpha amplitude and instantaneous frequency. We hope that these new analyses help to clarify how frequency and amplitude are interrelated and how these metrics are related to behavioral performance.

Note - changes to the text in response to reviewer comments are in blue in the all manuscript materials.

Reviewer #1 (Remarks to the Author):

The authors present a synchronization analysis of EEG signals studying the alpha band to infer mechanisms on behavioral performance. They mainly show that both amplitude and phase are interrelated there and one has to consider both. I am not a specialist in neuroscience and maybe this is an innovative result for that field.

We thank the reviewer for providing this summary and for making it apparent that our first submission did not adequately articulate the novelty of our main message and its theoretical importance to the field of sensory/cognitive neuroscience. We outline our arguments here and we edited the main text of the manuscript appropriately (with changes in the main text marked in blue).

With respect to the theoretical importance of the paper in sensory/cognitive neuroscience: there are two distinct theoretical camps that focus on either amplitude or phase/frequency as the putative mechanism by which alpha oscillations impact perception and behavior.

First, over the last 10-15 years, investigators have exerted a great deal of effort trying to understand how modulations of alpha amplitude are related to cortical inhibition and the

efficiency and selectivity of visual information processing. Several hundred peer-reviewed papers relating alpha amplitude with visual perception, attention, and behavior have been published over the last 10 years alone. For example, many studies have now shown that a shift of visual attention to one side of space is accompanied by a decrease in alpha amplitude over contralateral visual cortex. This observation has led to the prominent theory that decreases in alpha reflect a release from inhibition, thus improving visual processing, perception, and behavior¹⁻⁵.

Second, a separate line of studies has more recently focused on the link between changes in the phase and instantaneous frequency (IF) of alpha oscillations and behavior. These studies inspired the hypothesis that changes in phase and IF impact the effective sampling rate of visual perception. This idea was first introduced by studies showing that particular phases of the alpha oscillation are associated with improved detection performance^{6,7}. More recently, an elegant example was published in *Current Biology* suggesting that small changes in IF determine whether people will correctly perceive two rapidly presented flashes as distinct⁸.

Notably, studies that focus on either alpha amplitude or instantaneous frequency have essentially proceeded in parallel. We speculate that the lack of cross-talk occurs due to the assumption that amplitude and phase (and hence IF) are governed by independent variables. However, in a complex system, where oscillations with different characteristics are continuously interacting, we highlight that this assumption of independence is incorrect. Instead, as shown mathematically in Supplemental Material 1 and empirically in two separate data sets, the amplitude and frequency of a local oscillation are highly inter-dependent. This point needs to be made within basic neuroscience, using rigorous mathematical arguments and empirical observations, in order to begin to understand how alpha oscillations impact visual perception. Establishing the interdependence of these measures will be key in revising current theoretical frameworks toward a better understanding of the functional significance of large-scale cortical oscillations.

But from the complex systems science aspect (which I know much better) this phenomenon has been well understood and for basic systems it was clearly explained under which conditions a synchronization only among phases is possible and under which an interrelation of phases and amplitudes is necessary for reaching synchronization (cf. the review by Boccaletti et al., Phys. Rep. 2002 and references therein).

We thank the reviewer for pointing us to this important paper in which Boccaletti et al. demonstrate how chaotic systems synchronize under a variety of conditions. In relation to neural signals, the authors of that manuscript mention previous empirical results relating inter-areal phase synchronization of MEG and EEG signals to muscle activity and higher cognitive processes, highlighting the potential importance of coordinated oscillatory activity in information transfer between brain areas. However, their discussion focuses on how oscillators synchronize between areas (usually done by computing phase-phase and amplitude-amplitude correlation metrics) - not the relationship between amplitude and frequency per se. Thus, while the paper's treatment of different phase synchronization regimes is extremely important to understand how distinct neural populations synchronize to transfer information, it does not speak to how the changes in amplitude and frequency necessary for information processing in a given neural

population are yoked. This key distinction highlights an important gap in the field's understanding about how amplitude and frequency metrics are related. Our paper proposes a novel hypothesis and supporting evidence describing the relationship between amplitude and frequency in a given neuronal population, and does not speak to inter-areal synchronization.

Many high-profile papers have been published that examine amplitude or frequency without ever considering their interdependence, motivating the importance of investigating their relationship. This gap concerning amplitude-frequency interactions in the literature is critically important because the true dynamic properties of alpha oscillations are unknown. We believe that bringing ideas from complex-systems (in which the reviewer is clearly an expert) to bear on experimental observations will be key for developing more mathematically-grounded theories.

Finally, we also view the fact that the mathematical reasoning presented in our manuscript is well-supported in the reviewer's community as a positive endorsement of our overall approach. Given the rapid expansion of interest in this area, with hundreds of papers published each year focusing on the functional significance of alpha oscillations, our contribution will encourage a re-evaluation of empirical observations and existing theoretical frameworks.

Also the techniques for estimating phases and amplitudes used here are well known and were published several years ago (all of that is not cited in this contribution). There are also more advanced techniques on statistical tests for such estimates as applied here.

We cite well known work concerning the analysis of EEG signals, which has many considerations that other, more purely mathematical fields may not have (such as sampling rate, artifact presence, task induced non-stationarities, etc)⁹. We also use robust permutation based statistics that have many advantages over parametric approaches (i.e. fewer assumptions and more conservative estimates of significance). That said, we would be happy to consider any other methods or citations that the reviewer suggests.

Reviewer #2 (Remarks to the Author):

The study by Nelli et al represents a milestone for a better understanding of brain oscillations and alpha in particular. Particularly interesting - and novel - is the theoretical background as outlined in Fig.1. The prediction that correct trials are associated with an increase in (instantaneous) frequency and decrease in amplitude is well in line with the well documented observation that desynchronization in the upper alpha band is closely associated with the (cognitive) performance of a task. The article is well written, the experimental design and the results are convincing. I also want to say that the cited references represent an up to date review of relevant work. I have no concerns whatsoever.

The only thing that might improve the impact of the paper is a more detailed discussion of the results with respect to the length of the epochs (6 sec) on which the calculation of the spectra (for the LUT's) and the short time periods for the instantaneous and performance related changes in instantaneous frequency and Amplitude changes.

Response: Thank you for the supportive comments. Concerning the last point, this is indeed a key part of our framework. The use of averaging many long epochs to estimate amplitude spectra (as for our LUTs) is referred to as Welch's method. This method is common in spectral density estimation for achieving both 1) high frequency resolution and 2) low variance and stability in the estimate. Thus, by doing this we are estimating the stable amplitude spectra of each subject and electrode. Interestingly, as the reviewer notes, while amplitude and frequency fluctuate on very short time scales, the relationship between these fluctuations are captured through the amplitude spectra computed to emphasize stability. In fact, within a subject the three contralateral channels we analyzed are correlated at 0.97 ± 0.02 SD, illustrating that our technique tends to converge on similar, stable spectra for neighboring channels. In contrast, the three contralateral channels show a much weaker correlation of 0.69 ± 0.09 SD between subjects, confirming that these spectra are phenotypic and subject specific. We now discuss this in our Methods on page 18.

Reviewer #3 (Remarks to the Author):

Neural oscillations measured at the scalp with electroencephalography have been linked with a variety of aspects of perception and cognition. One of the dominant oscillations is in the alpha frequency band (~8-14 Hz). Alpha oscillations have been the focus of a lot of recent work and the submitted manuscript represents a timely contribution because it reveals an association between amplitude, instantaneous frequency, and behavior. More specifically, the authors show that both amplitude and instantaneous frequency are not only related to behavior, but fluctuations in frequency can predict amplitude on single trials. This contribution is important because a) amplitude and instantaneous frequency are often thought to be independent and b) it helps to link two explanations of the relationship between alpha oscillations and perception. The manuscript is extremely well-written and clear, the analyses are novel and include appropriate controls (especially using individual spectra to make LUTs), and the results are clear. There is no question in my mind that this work is appropriate for Nature Communications

I have only a few minor issues that should be pretty easy to address.

1. Studies that have shown a relationship between prestimulus oscillations and behavior, typically use a cue. Here, however, there is no cue and yet prestimulus increases in instantaneous alpha frequency on correct trials, but only contralateral to the upcoming stimulus. Given that participants couldn't predict where the stimulus would occur and there was only limited temporal predictability, this result seems a bit unexpected. Could the authors provide a bit more explanation? Does this change reflect some volitional process or is it more spontaneous?

Response: While we interpret this effect as likely reflecting an orienting of attention, whether this orienting is volitional or spontaneous is unknown given the lack of any explicit attention cues and the high degree of temporal/spatial uncertainty. Since we post-sort trials by accuracy, our ability to directly link behavioral effects with volitional deployments of attention is inherently limited (as is the case with any study that relies on post-sorting of trials based on behavior such as the classic Newsome random-dot studies that computed choice probabilities). Thus, the effects might reflect either spontaneous or volitional shifts of attention before the presentation of a stimulus. Importantly though, establishing the volitional nature of the effects was not the focus

of the current experiment – rather we focus on linking fluctuations in amplitude and instantaneous frequency to evaluate the relationship predicted by our mathematical analysis. While the lack of a cue makes the nature of these attentional fluctuations ambiguous, it also makes frequency and amplitude modulations uniquely attributable to the target event, instead of being potentially confounded with cue related activity.

We now discuss this important point raised by the reviewer in more details on page 7.

2. I understand the point of the analogy on p. 4. Given that it is not perfect, I'm wondering if there are a couple of steps that could be made to tighten it up a bit. For example, the "damping mechanisms" that constrain the amplitude of the jumps could be more explicitly linked with those that dampen neural oscillations.

Response: As described in the manuscript, dampening mechanisms such as the non-infinite stretchiness of a trampoline, constrain the amplitude of jumps. Similarly, transient (i.e. damped) neural oscillations are seen in response to stimuli. Damped oscillatory responses make neurons sensitive to fine temporal structure in input (be that sensory or activity induced by other neurons)¹⁰. Leak conductance, capacitance and voltage-gated currents are all mechanisms by which single neurons could uncouple from a rhythm, leading to damping of oscillatory amplitude¹¹. We now outline these possibilities in the text on page 4.

3. In the methods, the authors are pretty clear that they use a Butterworth bandpass when computing alpha amplitude. However, they also indicate that the data are also bandpassed during initial preprocessing, but it is unclear what was used for this bandpass.

Response: Thanks for catching this. This filtering was also done using a third order Butterworth filter. We have clarified this point in the main text on page 13.

Signed
Barry Giesbrecht

Reviewer #4 (Remarks to the Author):

In this paper, the authors attempted to closely examine the relationship between the instantaneous frequency and its amplitude of occipital alpha power. Although previous studies have highlighted the importance of each oscillatory signal in visual cognition, there hasn't been a systematic effort to link the two in a mechanistic manner. In that respect as well as the surging interest in utilizing and understanding EEG oscillatory signals as temporarily resolved signals of cognitive operations, I believe that such attempt would be of great impact and interest to the field. This being said, however, although I have no doubt this paper made an important first step in suggesting a case for the interdependence of the two oscillatory signals, I don't think that the current version of the manuscript is strong enough to warrant a publication at this journal. I would also like to emphasize that my concern can very well be resolved by additional analyses as well as re-phrasing of the manuscript.

The key finding in this manuscript is that by knowing the instantaneous frequency at a given moment, we can reliably predict the instantaneous amplitude of the occipital alpha power. This clearly demonstrates the interdependence of the instantaneous alpha amplitude on the instantaneous frequency, but when you examine the time course of instantaneous frequency and amplitude on correct v.s. incorrect trials, they do not exhibit the same time course. More precisely, the instantaneous alpha amplitude decreases after stimulus onset in a bilateral fashion with stronger suppression on the contralateral side to the stimulus. However, the effect of instantaneous frequency on correct v.s. incorrect trials is limited to a couple hundred milliseconds prior to the stimulus onset. Given these time course differences, it is very interesting and at the same time puzzling as to why the PaA generated by the look-up table analysis resulted in the pattern resembling the actual amplitude data. With regard to this puzzle, the authors do suggest an interesting hypothesis. The pre-trial shift in instantaneous alpha frequency represents the shift in the "sampling rate" of our visual system while post-stimulus shift in alpha amplitude (likely induced by both increase and decrease of alpha frequency, or pure decrease in amplitude unaccompanied by a frequency shift) represents efficient visual information processing. To potentially better answer this puzzle, I would like to see whether there is trial-type dependent differences in the post-stimulus shift in instantaneous frequency. One potentially interesting approach might be to classify trials based on the post-trial history of frequency shift and examine both amplitude shift as well as behavioral output. These analyses can potentially clarify the seemingly puzzling findings and provide additional insight into their interpretation of what instantaneous frequency and amplitude modulation reflect in terms of visual cognition.

Note that, on average, instantaneous frequency does continue to change after the stimulus. Since correlations between frequency and amplitude remain stable after the stimulus, this means changes in instantaneous frequency are related to those in amplitude. To understand this, it is important to remember that amplitude and frequency do not have a unique, one-to-one mapping. Instead, they are related by the non-monotonic "bump" shape of the amplitude spectra. This means that significant differences in one metric may "average out" in another metric. For example, amplitude could be equal at timepoints in which instantaneous frequency has moved from below to above peak alpha (or vice versa).

We also performed an additional analysis to further clarify how frequency and amplitude are related during the pre-stimulus and post-stimulus epochs during which frequency and amplitude (respectively) show accuracy-dependent effects. Trials were quartile binned based on average frequency in pre and post stimulus epochs based on figures 3 and 4 (much lower than peak alpha, lower than peak alpha, greater than peak alpha, or much greater than peak alpha). We then averaged pre or post-stimulus alpha amplitude and PaA over the trials in each of these trial bins. We observe a clear inverted-U relationship between amplitude and instantaneous frequency in both pre-stimulus and post-stimulus epochs (Figure 5e). This is consistent with the main analysis showing that each subject's alpha spectra could be used as a LUT to map changes in instantaneous frequency onto changes in alpha amplitude (see Figure 1).

Figure 5: Shifts in instantaneous frequency predict alpha amplitude

a) On a trial by trial and timepoint by timepoint basis, instantaneous frequency was used to generate predicted alpha amplitudes (PaA). PaA was baselined to the same interval used for alpha amplitude (-1000:-750 pre-target). Shaded areas indicate ± 1 SEM within subjects for correct (blue) and incorrect (red) trials. Reported results are averaged over the same groups of Contralateral and Ipsilateral electrodes previously reported. **b)** Correct – Incorrect differences are plotted for Contralateral and Ipsilateral electrodes. For illustration, topoplots indicate Correct – Incorrect topographies averaged over 100 ms bins centered on 400, 650 and 900 seconds after stimulus onset. All dots indicate significance from zero, evaluated by comparing the obtained t-value with a null distribution of t-values computed by shuffling the condition labels 10,000 times. This analysis was done on a timepoint-by-timepoint basis from stimulus onset to +1000 ms, as indicated by the non-shaded areas (see Methods). Main effects of accuracy indicated in blue and yellow in contralateral and ipsilateral electrodes, red indicates a main effect of topography. Gray dots indicate significance after FDR correction at $P < 0.05$. **c)** To assess how well PaA corresponds with alpha amplitude, we computed correlation values for each subject and electrode for each timepoint over the entire -750 to 1000 ms peri-stimulus interval. Histograms show correlation values concatenated over all subjects, timepoints and electrodes on correct (blue) and incorrect (red) trials in the contralateral and ipsilateral electrodes. Stars on each panel indicate that all correlation values shown in these histograms are significantly different from correlations obtained with shuffled LUTs (see Methods, Supp Figure 4). Dots in histograms indicate the median correlations for that condition. Timecourse panels on the right show these correlations are relatively stable over time, where the y axes of the plots run from 0.425 to 0.525, corresponding to the gray shaded area in the histogram. All dots indicate significant difference in the correlations between conditions, evaluated by comparing the obtained t-value with a null distribution of t-values computed by shuffling the condition labels 10,000 times. This analysis was done on a timepoint-by-timepoint basis from -500 to +1000 ms, as indicated by the non-shaded areas. Main effects of are in black, while

purple indicates an interaction. Gray dots indicate significance after FDR correction at 0.05. **d)** Three example trials from three different subjects show how PaA shifts during single trials mirror those in alpha amplitude. The y-axis and traces for PaA are indicated in dashed lines, while those for amplitude are solid. Note that the y-axis range is different in each subplot to maximize visibility of amplitude and PaA (see Methods). **e)** Trials were sorted according to mean pre (-350:-50 ms) and post (350:650 ms) stimulus frequency. Average amplitude and PaA were then computed on these trials and time points. Trials were further split by accuracy, as indicated by blue and red lines. Significant differences were computed using a two way repeated measures ANOVA. Black stars indicate a significant effect of frequency, while a purple star indicates a significant effect of accuracy in the two-way ANOVA.

Table 5: The effect of frequency bin and accuracy on mean amplitude and PaA in pre and post stimulus epochs from -350:-50 ms or 350:650 ms relative to the stimulus (timepoints chosen according to the significant timepoints from figures 3 and 4). Trials were binned by their average pre or post stimulus frequency, and then amplitude and PaA was computed for each of these bins. We then computed a 2 way repeated measures ANOVA with frequency and accuracy as factors was computed for these values. F-values were compared against distributions obtained empirically by randomizing condition labels 10,000 times and then repeating the same statistical test (see Methods). * indicates that p-values were significant alpha = 0.05. All statistics are reported for contralateral channels, as plotted in figure 5e.

	Amplitude		PaA	
	Prestimulus	Poststimulus	Prestimulus	Poststimulus
Frequency (Bins 1-4)	*F(3, 15) = 16.9, p=0.0	*F(3, 15) =16.9, p=0.0008	*F(3,15) = 28.7, p = 0.0	*F(3, 15) =28.6, p=0.0008
Accuracy (corr vs incorr)	n.s.	*F(1, 15) = 9.2, p=0.0028	n.s.	n.s.
Freq*Acc interaction	n.s.	n.s.	n.s.	n.s.

Another concern that I had was the generalizability of the findings. The authors did report the similar interdependence of the instantaneous frequency and the amplitude modulation in two different experiments, but their analyses for Experiment 2 seemed incomplete. For instance, what was the nature of the time course of instantaneous frequency effect and amplitude effect? Were they more or less consistent with what they found in Experiment 1 (i.e. pre-stimulus increase in contralateral instantaneous frequency for correct trials and post-stimulus bilateral amplitude suppression (larger for contralateral channels) for correct trials)? The readers would benefit a lot from knowing these details in judging not only the consistency of the findings as well as the validity of the authors' interpretation as to what cognitive mechanisms the instantaneous frequency and amplitude effect might reflect.

Our original focus in including Experiment 2 was to independently verify the *relationship* between amplitude and frequency, not necessarily to verify what previous reports have found (i.e. an increase in instantaneous frequency and a post-stimulus drop in amplitude on correct trials). As such, we used data from the 2nd experiment to focus on replicating the PaA-Amplitude correlations observed in the first experiment. Additionally, there are many aspects of the

experimental design of the previously-published study that make it unsuitable for an analysis of the relationship between instantaneous frequency and behavior as we show in E1 (and as shown by ⁸).

First, and most important theoretically, is that the “sampling rate” interpretation of frequency isn’t readily applicable to the second experiment. This is because 1) the targets were presented for 300 ms, or ~3 full alpha cycles, lessening the need for denser temporal sampling and 2) the experiment required subjects to employ working memory to compare the two stimuli.

Furthermore, 3 visual stimuli (a cue and two jittered targets) were presented during each trial. Each of these alter the timecourse of instantaneous frequency and amplitude, as shown in Figure 6. As target presentations were jittered relative to the cue, averaging over trials leads to a loss in interpretability and statistical power. Each of the event related changes may interact with both each other and the endogenous signal, and thus the experiment does not offer the same ability to uniquely attribute frequency and amplitude shifts to a single event.

Lastly, a wide range of contrast levels, focused vs. divided attentional conditions, and a low number of incorrect trials (76.1 +/- 0.6% SEM accuracy) further complicate assessing a direct analog of the first experiment with appropriate statistical power.

With these caveats in mind, we now plot out these timecourses and perform statistics for the divided attention trials (most analogous to the previous experiment) as shown below. We have added these figures to the supplementary materials of the main paper as well. In agreement with the first study presented, these timecourses exhibit a significant interaction between topography and accuracy on post-stimulus alpha amplitude. Additionally, instantaneous frequency before stimulus 1 increased on correct and low contrast trials, a result we argue is most analogous with the high spatial and stimulus uncertainty subjects experienced in the first experiment (i.e., subjects did not know ahead of time where the stimulus would appear and stimuli were rendered at very low contrast). Thus, the results of these analyses in Experiment 2 are generally consistent with the pattern of effects that we report in Experiment 1.

Supp 5: Amplitude (left panel) and frequency (right panel) locked to the onset of the first stimulus (top half) and the onset of the second stimulus (bottom half). All plots are during the divided attention condition, and are shown as a function of topography (contralateral and ipsilateral) and behavioral performance (correct in blue, incorrect in red). Results of a 3 way repeated measures ANOVA analysis with topography, accuracy and contrast level are plotted at the bottom of each subplot indicating uncorrected p-values < 0.05 with colors corresponding to the legend, while effects that survive FDR correction are in black (full results of this analysis are described in Tables S5a and S5b). Post-hoc t-tests were performed between p correct and incorrect timecourses to further understand these effects. Timepoints with $p < 0.05$ from these t-tests are plotted at the top of each subplot in black, with timepoints that survive FDR correction in cyan (contralateral) or yellow (ipsilateral). In keeping with previous figures, all amplitude statistics were performed on post-onset timepoints, while frequency statistics were performed on pre-onset timepoints. Finally, all p-values were determined by randomizing conditions 5,000 times and comparing observed values to these empirical distributions. 0

	Post Stimulus 1 Amplitude	Post Stimulus 2 Amplitude
Topography (contralateral vs ipsilateral)	4.39 < F(1,14) < 6.73 0.02 ≤ p < 0.05	n.s.
Accuracy (correct vs incorrect)	n.s.	4.24 < F(1,14) < 5.60 0.026 ≤ p < 0.05
Contrast (levels 1- 3)	3.23 < F(1,14) < 4.87 0.013 ≤ p < 0.05	3.19 < F(1,14) < 5.18 0.008 ≤ p < 0.05
Topography x Accuracy	4.46 < F(1,14) < 10.71 *0.005 ≤ p < 0.05	4.44 < F(1,14) < 15.40 *0 ≤ p < 0.054.4
Topography x Contrast	n.s.	n.s.
Accuracy x Contrast	n.s.	3.50 < F(1,14) < 3.65 0.047 ≤ p < 0.05
Topography x Accuracy x Contrast	n.s.	n.s.

Supp 5, Table A: Amplitude during the divided attention condition as a function of electrode location and behavioral performance. All tests report the maximum and minimum timepoint-by-timepoint F-values over poststimulus timepoints. F-values were compared against distributions obtained empirically by randomizing condition labels 5,000 times and then repeating the same statistical test (see Methods). * indicates that p-values were significant after FDR correction at alpha = 0.05 from stimulus onset to +750ms. The minimum uncorrected p-value is also reported for each interval (note all intervals have a maximum uncorrected p-value of 0.05 since they were chosen on this basis).

	Pre Stimulus 1 Frequency	Pre Stimulus 2 Frequency
Topography (contralateral vs ipsilateral)	4.6 < F(1, 14) < 8.44 0.009 ≤ p < 0.05	n.s.
Accuracy (correct vs incorrect)	4.41 < F(1, 14) < 4.42 0.048 ≤ p < 0.05	n.s.
Contrast (levels 1-3)	3.31 < F(1, 14) < 3.70 0.030 ≤ p < 0.05	n.s.
Topography x Accuracy	n.s.	n.s.
Topography x Contrast	3.37 < F(1, 14) < 4.01 0.031 ≤ p < 0.05	n.s.
Accuracy x Contrast	3.31 < F(1, 14) < 10.44 *0 ≤ p < 0.05	n.s.
Topography x Accuracy x Contrast	n.s.	n.s.

Supp 5, Table B: Frequency during the divided attention condition as a function of electrode location and behavioral performance. All tests report the maximum and minimum timepoint-by-timepoint F-values over prestimulus timepoints. F-values were compared against distributions obtained empirically by randomizing condition labels 5,000 times and then repeating the same statistical test (see Methods). * indicates that p-values were significant after FDR correction at alpha = 0.05 from -500 to stimulus onset. The minimum uncorrected p-value is also reported for each interval (note all intervals have a maximum uncorrected p-value of 0.05 since they were chosen on this basis).

Taken together, although I have no doubt that this paper will make a great impact to the field, I think that the current manuscript would benefit a lot from a major revision.

Minor points.

The baseline window for the amplitude analyses was set to -1000 to -750ms before the stimulus onset. Was there a specific reason why this window was chosen?

We chose a baseline that would not conflate any analysis of pre-stimulus activity, which is seen in other experiments in epochs starting roughly 400 ms before the stimulus (for example see ^{6,8}). Thus, to keep the time-epoch used for baselining free from any time frequency effects, we chose to baseline trials from this epoch well before this prestimulus period. However, as long as the baseline is before the -400ms epoch in which pre-stimulus effects are commonly seen, the choice does not matter for our results.

Figure 4.

Although I was able to figure out the color-coding for panel A thanks to its consistency with other figures, it is a good idea to put a legend on every panel.

We did this, thank you.

Page 14

The first sentence of Instantaneous Frequency section.

Please capitalize the first "i" in the "instantaneous".

We did this, thank you.

Thank you for your consideration, and please let us know if you have any questions concerning our submission.

References

1. Sauseng, P. *et al.* A shift of visual spatial attention is selectively associated with human EEG alpha activity. *Eur J Neurosci* **22**, 2917–2926 (2005).
2. Klimesch, W., Sauseng, P. & Hanslmayr, S. EEG alpha oscillations: the inhibition-timing hypothesis. *Brain Res. Rev.* **53**, 63–88 (2007).
3. Fries, P., Womelsdorf, T., Oostenveld, R. & Desimone, R. The effects of visual stimulation and selective visual attention on rhythmic neuronal synchronization in macaque area V4. *J. Neurosci.* **28**, 4823–4835 (2008).
4. Rihs, T. A., Michel, C. M. & Thut, G. Mechanisms of selective inhibition in visual spatial attention are indexed by α -band EEG synchronization. *Eur. J. Neurosci.* **25**, 603–610 (2007).
5. Haegens, S., Nácher, V., Luna, R., Romo, R. & Jensen, O. α -Oscillations in the monkey sensorimotor network influence discrimination performance by rhythmical inhibition of neuronal spiking. *Proc. Natl. Acad. Sci. U. S. A.* **108**, 19377–82 (2011).
6. Busch, N. a, Dubois, J. & VanRullen, R. The phase of ongoing EEG oscillations predicts visual perception. *J. Neurosci.* **29**, 7869–7876 (2009).
7. Mathewson, K. E., Gratton, G., Fabiani, M., Beck, D. M. & Ro, T. To see or not to see: prestimulus alpha phase predicts visual awareness. *J. Neurosci.* **29**, 2725–2732 (2009).
8. Samaha, J. & Postle, B. R. The Speed of Alpha-Band Oscillations Predicts the Temporal Resolution of Visual Perception. *Curr. Biol.* **25**, 2985–2990 (2015).
9. Cohen, M. X. Fluctuations in oscillation frequency control spike timing and coordinate neural networks. *J. Neurosci.* **34**, 8988–98 (2014).
10. Izhikevich, E. M. Resonate-and-fire neurons. *Neural Networks* **14**, 883–894 (2001).
11. Buzsáki, G., Andreas, D. & Draguhn, A. Neuronal Oscillations in Cortical Networks. *Science (80-)*. **304**, 1926 (2004).

REVIEWERS' COMMENTS:

Reviewer #3 (Remarks to the Author):

The authors have submitted a revision that addresses each of my previous concerns. In addition, based on my reading, the authors have thoughtfully addressed the thorough and constructive comments of the other reviewers.

Reviewer #4 (Remarks to the Author):

I believe that the authors did a fantastic job responding to my concerns. The current manuscript will be a foundational paper for the future studies examining the oscillatory mechanisms underlying cognition.